

# Carbon burial in deep-sea sediment and implications for oceanic inventories of carbon and alkalinity over the last glacial cycle

Olivier Cartapanis[1,2], Eric D. Galbraith[2,3,4], Daniele Bianchi[5], and Samuel L. Jaccard[1]

1. Institute of Geological Sciences and Oeschger Centre for Climate Change Research, University of Bern, 3012 Bern, Switzerland;
2. Earth and Planetary Sciences McGill University, Montreal H3A 2A7, Canada.
3. Institució Catalana de Recerca i Estudis Avançats (ICREA), Pg. Lluís Companys 23, 08010 Barcelona, Spain
4. Institut de Ciència i Tecnologia Ambientals (ICTA) and Department of Mathematics, Universitat Autònoma de Barcelona, 08193 Barcelona, Spain
5. Department of Atmospheric and Oceanic Sciences, University of California Los Angeles, Los Angeles, CA 90095-1565, USA

*Correspondence to*: Olivier Cartapanis (olivier.cartapanis@geo.unibe.ch)

**Abstract.** Although long assumed that the glacial-interglacial cycles of atmospheric $CO_2$ occurred despite a constant 'active' carbon inventory, there are signs that the geological $CO_2$ supply rates varied. However, changes of the carbon inventory

cannot be assessed without constraining the removal rates from the system, which mainly occurs in marine sediments. Here, we present the first global reconstruction of carbon and alkalinity burial in deep-sea sediments over the last glacial cycle. Although subject to large uncertainties, the reconstruction provides a first order constraint on changes in carbon and alkalinity inventories over the last glacial cycle. The results suggest that reduced burial of carbonate in the Atlantic Ocean was not entirely compensated by the increased burial in the Pacific basin during the last glacial period. The burial-driven

inventory variations are likely to have significantly altered the $\delta^{13}C$ of the ocean-atmosphere carbon, as well as the DIC and alkalinity budget, confirming that the active carbon inventory was a dynamic, interactive component of the glacial cycles.





## 1. Introduction

On geological timescales, the burial of carbon in marine sediments removes it from the climatically 'active' inventory of the ocean and atmosphere (Broecker, 1982; Opdyke and Walker, 1992; Sigman and Boyle, 2000; Wallmann et al., 2015) (Fig. 1). Dissolved inorganic carbon (DIC) is extracted from seawater by photosynthetic organisms and transformed into organic

5  carbon (Corg), while calcifying organisms use DIC to produce carbonate minerals ($CaCO_3$). Most of the Corg and a large fraction of the CaCO3 is then recycled within the ocean, but a small portion escapes to be buried at the seafloor, in both coastal and pelagic environments. While Corg burial only removes carbon from the ocean, $CaCO_3$ burial removes alkalinity (ALK) as well as carbon, which decreases the solubility of $CO_2$ in seawater and thereby tends to shift carbon from the ocean to the atmosphere.

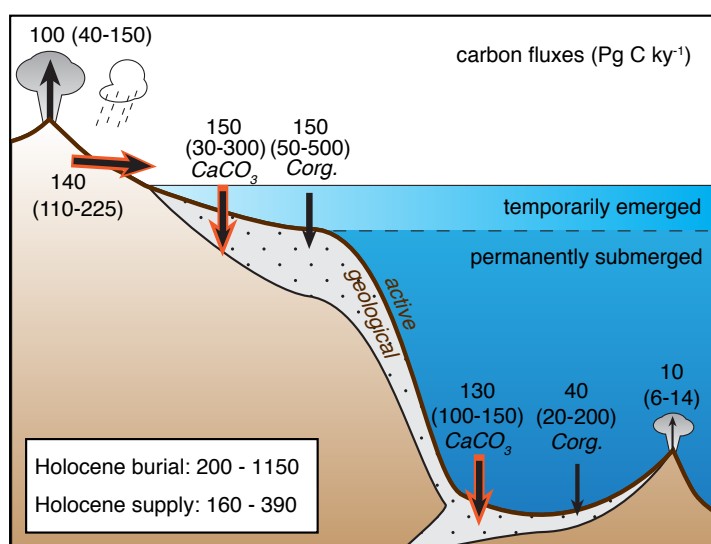

**Figure 1: Estimated modern global transfers of carbon between the geological and active inventories. Sea level variations on glacial-interglacial timescales distinguish two fundamentally different domains of seafloor on which carbon burial has occurred: surfaces fringing the continents that emerged as ice sheets grew, and the permanently-submerged surfaces of continental slopes and the deep sea. As discussed in section 2, the modern fluxes are highly**

15  **uncertain. Ranges in parentheses show the values found in the literature, above which representative modal values are shown. The total carbon inventory in the pre-industrial ocean, atmosphere and biosphere is estimated to have been ~41,000 PgC, 93% of which was located in the ocean (Hain et al., 2014a).**



Because these marine burial fluxes dominate the removal of carbon from the active inventory and thus affect the air-sea partitioning of $CO_2$ via alkalinity, they place a first-order control on the long-term carbon cycle. The burial of $CaCO_3$ in the deep sea is thought to play a pivotal role by responding to changes in the deep ocean $CO_3^{2-}$ concentration (approximately equal to ALK-DIC) with enhanced dissolution or preservation, adjusting the deep ocean ALK to 'compensate' for the

perturbation with an e-folding timescale of a few thousand years (Broecker and Peng, 1987), but the details of this process remain poorly quantified. Anthropogenic carbon emissions are significantly altering the marine carbon cycle and air-sea partitioning by injecting carbon into the system very rapidly, with respect to natural changes (Orr et al., 2005; Hoegh-Guldberg et al., 2007; Keil, 2017). Despite their much slower rates, natural changes of the recent geological past can help inform basic process understanding of the carbon cycle, and thereby contribute to reducing uncertainties in long-term future

climate projections (Schmittner et al., 2007).

Several mechanisms that have been proposed to explain low glacial atmospheric $CO_2$ concentrations involve decreased $CaCO_3$ burial (Vecsei and Berger, 2004; Hain et al., 2014a), which would have raised ALK and thereby increased $CO_2$ solubility. One of these calls on changes in ocean circulation (Boyle, 1988; Jaccard et al., 2009) or biological production (Archer et al., 2000; Omta et al., 2013) to have raised the DIC in the deep ocean and therefore have lowered deep ocean

$CO_3^{2-}$, requiring the ocean to compensate by dissolving $CaCO_3$ until the alkalinity was raised sufficiently to restore $CO_3^{2-}$ and re-balance burial with the alkalinity supplied by weathering (Broecker and Peng, 1987; Yu et al., 2014). Another proposal is that the rapid burial of $CaCO_3$ in coral reefs and carbonate platforms would have been curtailed by falling sea level as ice sheets grew (Berger, 1982; Opdyke and Walker, 1992; Vecsei and Berger, 2004; Wallmann et al., 2015), although it has been suggested that this would have been compensated by opposite changes in the deep sea (Opdyke and

Walker, 1992; Roth and Joos, 2012) and therefore could have had a minor overall impact. Meanwhile, it is typically assumed that the active carbon inventory was constant over glacial cycles in most carbon budgeting exercises (Sigman and Boyle, 2000; Hain et al., 2014b), and although it has been suggested that geological carbon inputs contributed to the deglacial rise of $CO_2$ (Marchitto et al., 2007; Huybers and Langmuir, 2009; Stott et al., 2009; Ciais et al., 2012; Zech, 2012; Lund et al., 2016), any such input could have been buffered by changes in deep-sea carbonate burial and the resulting

feedback on ocean alkalinity.

Although a number of modeling studies have considered these inventory-altering mechanisms (Opdyke and Walker, 1992; Wallmann et al., 2015), they are often considered in isolation, and quantitative reconstruction of the entwined controls on the carbon and alkalinity inventories has been hampered by a lack of observational constraints on the four burial fluxes shown in Fig. 1. Recently, a reconstruction of global deep-sea burial of Corg was reported (Cartapanis et al., 2016), providing a record

for one of these fluxes. Here, we present a new reconstruction of global deep sea (deeper than $\approx$ 200 m) burial of $CaCO_3$, completing the deep sea budgets of carbon and alkalinity burial. This reconstruction constrains the net result of carbonate compensation and allows some first-order expectations for changes in carbon and alkalinity inventories to be illustrated. These expectations show a high likelihood that large change in both carbon and alkalinity inventories would have been



driven by changes in burial, and highlight the potentially important role of variations in shelf burial fluxes, which remain poorly constrained.

## 2. Natural sinks and sources of carbon in the climatic system

The removal of dissolved inorganic carbon (DIC) from the ocean occurs via two very different phases, organic carbon and

calcium carbonate minerals, with distinct responses to environmental conditions. Meanwhile, carbon input to the climatic system can be summarized as two sources: geological emissions of carbon dioxide (from igneous and metamorphic sources), and weathering of carbon-bearing sediments on continental surfaces.  Here we review each of these source and sink terms, including estimates of their respective mass fluxes and carbon stable isotopic composition.

### 2.1 Carbonate burial in oceanic sediment

Benthic organisms are the main producers of carbonates in warm water reef and carbonate platform ecosystems. Estimating the global burial rates of carbonate in the heterogeneous coastal environments of the world has proven a challenge. According to Vecsei and Berger (2004), the carbonate buried in coral reef environments during the last 6 kyrs was equivalent to 28 PgC/kyr, and had been twice as high during the early Holocene (50 PgC/kyr). Earlier, it had been estimated that the volume of carbonates associated with Holocene reefs was much larger than this, with 168-228 PgC/kyr burial over the last 5

kyr, and an additional 53 PgC/kyr burial occurring over shallow water carbonate platforms (Opdyke and Walker, 1992). An intermediate value for late Holocene reefs was reported by (Ryan et al., 2001) who suggested that there was a twofold decrease of total inorganic carbon burial from the early- to the late Holocene, with average values for the Holocene of 156 PgC/kyr. Using a different approach, based on the carbon budget of coastal zones, Bauer et al. (2013) estimated inorganic carbon burial in coastal sediments as 150 PgC/kyr, consistent with the Ryan et al. (2001) estimates. A similar figure was also

reached by (Milliman, 1993), who estimated that half of the global carbonate burial (180 out of 360 PgC/kyr) occurs in shallow water environment. Thus, even though these studies use different approaches, and do not cover exactly the same depositional settings and temporal intervals, the estimates for modern inorganic carbon burial in shallow water environments converge to an average value of 150 PgC/kyr (ranging between 30 and 300 PgC/kyr).

In open ocean sediments, carbonates are mostly associated with coccolithophorid and foraminiferal remains. The burial of

carbonates in deep-sea sediment is controlled by the production and export of carbonate minerals from the surface ocean on one hand, and by the preservation of these phases at the seafloor on the other (Dunne et al., 2012). The carbonate saturation state increases with the *in situ* concentration of $Ca^{2+}$ and $CO_3^{2-}$, and decreases with increasing pressure, thus with water depth. The accumulation of respired carbon in deep water along the pathway of the global ocean circulation tends to acidify more poorly-ventilated parts of the deep ocean, thereby reducing carbonate preservation with water mass aging as shown in

Fig. 2. Thus, carbonate burial in the modern environment is high in the Atlantic, where the carbonate compensation depth



(CCD) is relatively deep, in the eastern equatorial Pacific, where production is high, and in the southern Indian Ocean where high productivity coincides with elevated bathymetric features.

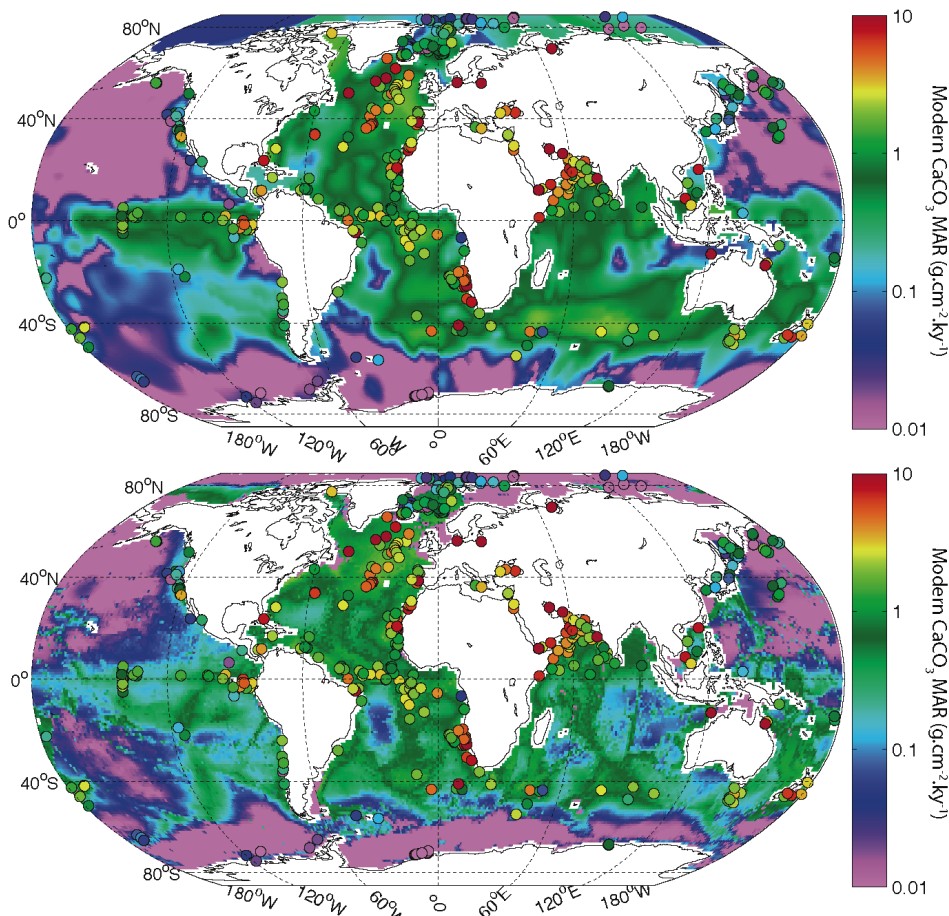

**Figure 2: Top. CaCO₃ burial based on sedimentary data; Bottom. CaCO₃ burial based on a metamodel optimized to fit the sedimentary data (Dunne et al., 2012). Dots correspond to Holocene CaCO₃ burial from sediment cores used in the reconstruction of past burial fluxes. Note that many sediment cores were collected at sites with accumulation rates that are higher than the local average sedimentation rate, a common practice in paleoceanography (see appendices).**





By contrast, the deep north Pacific, bathed by aged, DIC-rich waters, shows very low carbonate burial rates. The fraction of pelagic $CaCO_3$ exported from the surface ocean that is eventually buried in sediments is estimated to vary between 0.6 and 0.06, depending on surface export, bottom water condition, organic carbon respiration and lithogenic flux (Dunne et al.,
5 2012).

Based on modern surface sediment composition and estimates of bulk sediment burial rate, and also considering the export flux of $CaCO_3$ from the surface ocean, the modern $CaCO_3$ burial in deep sea environments has been estimated to be between 100 and 130 PgC/kyr (Catubig et al., 1998; Sarmiento and Gruber, 2006; Dunne et al., 2012). Up to 10 % of the carbonates buried in deep sea sediment could have been authigenically precipitated (Sun and Turchyn, 2014). Thus, modern estimates
suggest that carbonate burial is roughly equally distributed between shallow- and deep-sea environments, but with significant uncertainty in their ratio (at least a factor of 2).

The globally-averaged carbon isotopic signature ($\delta^{13}C$) of carbonate buried in marine sediment is surprisingly poorly-constrained. While the $\delta^{13}C$ composition of marine carbonates has been studied intensively, those studies generally consider some specific fractions of the carbonate material, such as benthic or planktic foraminifers, while the $\delta^{13}C$ of the bulk
carbonate is only rarely reported. However, it is often assumed that carbonate precipitates with an isotopic composition close to that of the ambient dissolved inorganic carbon, while some modeling experiments suggests a fractionation of 0.78 ‰ as compared to ambient DIC (Kump and Arthur, 1999).

### 2.2 Organic carbon burial in marine sediment

The organic carbon buried in marine sediments is supplied by photosynthetic carbon fixation by marine phytoplankton and
terrestrial plants. Marine organic carbon production by phytoplankton in the surface ocean amounts to ~45 PgC/yr in pelagic ecosystems (92 % of the total ocean surface), whereas ~9 PgC/yr is produced in coastal ecosystems (8 % of the total ocean surface, continental margins being defined here as regions shallower than 1,000 m) (Sarmiento and Gruber, 2006) and references therein. However, only a minor fraction of the organic matter produced in the sunlit surface ocean reaches the sediment before being respired by heterotrophs (0.34 PgC/yr for open ocean settings and 2.5 PgC/yr for continental margins
(Sarmiento et al., 2002)). Because organic matter is further consumed by benthic organisms on the sea floor and in the uppermost layers of the sediment, very little is permanently buried in sediments, with the highest burial fractions occurring in rapidly-accumulating anoxic sediments on continental margins (Sarmiento and Gruber, 2006). Meanwhile, terrestrial organic matter may represent up to a third of the total organic matter buried in marine sediment, with the majority of terrestrial organic material burial occurring in deltaic and continental margin sediment (Burdige, 2005).
The available estimates for modern burial of organic carbon ($C_{org}$) show large ranges. As shown by (Burdige, 2007), global $C_{org}$ burial estimates vary by more than one order of magnitude (50-2600 PgC/kyr), with several estimates significantly higher than the commonly cited range of 120-210 PgC/kyr (Berner, 1989; Hedges and Keil, 1995; Sarmiento and Gruber, 2006). These discrepancies are probably linked to an incomplete understanding of the sedimentary environments in coastal





regions, and to the different integration time of the measurements used to determine global burial on shelves (Burdige, 2007). Attempting to account for these weaknesses, Burdige (2007) suggests a total global $C_{org}$ burial flux of 310 PgC/kyr.

By contrast with carbonate burial that shows relatively similar burial rates over deep-sea sediment and shallow environments, studies have generally shown that $C_{org}$ burial is strongly focused over continental shelves. The reason for this

contrast is that heterotrophic organisms rapidly consume organic matter, so that sinking fluxes decrease sharply with water depth, whereas carbonate minerals sink relatively unimpeded through the water column. For example, according to the estimates of (Dunne et al., 2007), burial of carbon is 480 PgC/kyr in near-shore deposits (< 50 m), 190 PgC/kyr on shelves (50-200 m), and 100 PgC/kyr on slopes (200-2000 m), greatly exceeding the burial of 12 PgC/kyr on deep-sea rises and plains, even though the deep-sea sediment covers nearly 90 % of the ocean floor. Burdige (2007) estimated that burial of 250

PgC/kyr occurs on continental margins < 2,000 m water depth, with a remaining 60 Pg C/kyr in deeper waters. We note that Muller-Karger et al. (2005) estimated that around 40 % of the total burial occurs in margin sediment between 50 m and 2,000 m (ignoring the shallowest 50 m), while the rest is buried in deep-sea sediment, which represents a much greater deep-sea burial proportion than other studies.

Because carbon stable isotope fractionation occurs during photosynthesis, organic matter $\delta^{13}C$ is markedly different than the

carbon pool from which it was formed (Degens, 1969; Freeman and Hayes, 1992). $\delta^{13}C$ of organic matter in modern marine surface sediment reflects a variety of factors, and ranges between -20 and -22.5 ‰ (Wang and Druffel, 2001). Using our database, we extracted all the $\delta^{13}C$ measurements performed on marine sediment organic matter within the uppermost 10 cm of the sediment. We estimated the mean $\delta^{13}C$ of organic matter in the first 10 cm of the sediment at -22.2 ‰ with a standard deviation of 2.3 ‰, consistent with the literature (Sundquist and Visser, 2003).

**2.3 Geological $CO_2$ emissions**

Modern estimates of volcanic emission of $CO_2$ range between 6 and 14 PgC/kyr for mid oceanic ridges alone (Chavrit et al., 2014) and 44 PgC/kyr for mid-oceanic ridges plus volcanic arcs (Coltice et al., 2004). Estimated emissions on geological timescales are notably higher, ranging from 40 to 150 PgC/kyr (Sundquist and Visser, 2003; Burton et al., 2013), consistent with relatively quiet volcanic activity during the late Holocene (Gerlach, 2011). It has been suggested that increased

emission during deglaciations, owing to mantle decompression caused by ablation of glaciers and ice caps could have reached 200 PgC/kyr (Huybers and Langmuir, 2009; Roth and Joos, 2012). Continental rifting zones could also represent a significant source of exogenic carbon to the climatic system, amounting to 14 to 26 PgC/kyr, (Lee et al., 2016). Notably, a recent study significantly increased the potential contribution of solid earth outgassing as compared to previous estimations (see details in Burton et al. (2013) and references therein), a revision that highlights the large degree of uncertainty in this

important flux.

Some reconstructions for past global volcanic activity have suggested a tight coupling between glacial ice volume and volcanic eruption frequency (Jellinek et al., 2004). Increased eruption frequency among subaerial volcanoes appears to have





occurred following the highest rate of sea level rise (with a 4 kyr lag), presumably in response to mass unloading and isostatic adjustment (Huybers and Langmuir, 2009; Kutterolf et al., 2012; Roth and Joos, 2012). Subsea volcanism at mid ocean ridges also appears to vary in response to isostatic loading as ice sheets grow and sea level drops. Lund et al. (2016) suggested that the maximum rates of $CO_2$ emission could occur approximately 10 ky after a sharp sea level drop, which

occurred most recently at the start of the last deglaciation. The isotopic composition of volcanic emissions has been estimated at around -5 ‰ (Kump and Arthur, 1999; Cartigny et al., 2001; Deines, 2002; Roth and Joos, 2012).

**2.4 Erosion of continental surfaces.**

The second main source of carbon to the ocean-atmosphere system consists in the transfer of carbon from exposed rocks through physical and chemical weathering. Carbonate rock weathering currently adds between 70 PgC/kyr (Hartmann et al.,

2014) and 155 PgC/kyr (Gaillardet et al., 1999) to the ocean/atmosphere system, while the petrogenic particulate organic carbon flux to the ocean is thought to amount to between 40 PgC/kyr (Copard et al., 2007; Galy et al., 2015) and 70 PgC/kyr (Petsch, 2014). Alkalinity fluxes from the continent to the ocean also impact the carbon system. Most of the alkalinity released from continental surface corresponds to the weathering of carbonate (63 to 74 %, (Gaillardet et al., 1999)), with the remainder supplied by silicate weathering. Total alkalinity fluxes to the ocean amount to between 30 Pmol equ/kyr (Amiotte

Suchet et al., 2003) and 40 Pmol equ/kyr (Gaillardet et al., 1999).

The degree to which weathering varied over glacial-interglacial cycles is highly uncertain. An early modelling study estimated that global chemical erosion was 20 % higher during the LGM (Gibbs and Kump, 1994), mostly because of enhanced erosion of newly exposed continental shelves. However, Vance et al. (2009) have argued that silicate chemical weathering was much lower during glacial. It is commonly assumed that physical erosion of continental surfaces was higher

during glacial because of enhanced action of ice sheets at high latitude (Herman et al., 2013) and the exposure of shelves, although this too has been debated (von Blanckenburg et al., 2015), while it is likely that chemical weathering could have been reduced due to lower temperature so that global erosion remained relatively stable over glacial interglacial timescales (Foster and Vance, 2006).

The isotopic composition of the global erosion flux depends on the balance of organic carbon to carbonate weathering, and

has been estimated to range between -8.5 ‰ and -5 ‰ (Kump and Arthur, 1999; Roth and Joos, 2012).



### 3. Material and method: Global carbonate burial reconstruction

#### 3.1 Analytical strategy overview

Sediment cores sample only a tiny fraction of the global seafloor. In order to use these unevenly distributed observations to infer integrated changes in global carbon burial, we need to take into account the following aspects: (1) the limited spatial

extent that is integrated by each core; (2) the fact that accumulation rates at an individual site may not be representative of the accumulation within a broader region; and (3) the fact that carbonate accumulation rates vary widely between different parts of the world.

To account for these potential shortcomings, our quantification of global carbonate burial includes 3 elements. (1) An estimate of the spatial variations in global carbonate mass accumulation rate (MAR) in the modern ocean derived from

sediment core tops and process modeling (Dunne et al., 2012); (2) A subdivision of the ocean into coherent biogeochemical provinces, in order to account for the spatial and bathymetric heterogeneities in the sedimentary carbonate burial record. Because they take into account spatial structure in carbonate production and preservation, the province approach is preferable to simple mapping techniques, which are also prone to interpolation and extrapolation biases, especially in regions where measurements are scarce (see details in Appendix A); (3) Reconstruction of the relative variation of the carbonate

MAR in each province over time, based on a compilation of available marine sediment proxy records. These relative changes are then multiplied by the modern $CaCO_3$ MAR estimate in each province, and integrated to assess the global history of deep-sea carbonate burial (see also (Cartapanis et al., 2016)). We note that the last step implicitly assume that carbonate currently present in modern sediment is in equilibrium with its surrounding environment, and is not being gradually dissolved (Berelson et al., 2007) nor precipitated (Sun and Turchyn, 2014), which would be expected to alter modern $CaCO_3$

MAR to some degree.

#### 3.2 Modern carbonate MAR map

We use two different, but closely related maps of modern carbonate MAR (Fig. 2), both of which were previously published by Dunne et al. (2012). The first is based only on observations and was determined by multiplying the carbonate concentration in surface sediments by the bulk sediment accumulation rate. The carbonate concentration map was generated

by combining several literature sources (Seiter et al., 2004; Dunne et al., 2007), whereas the bulk MAR map was determined based on the geometric mean of two preexisting maps (Jahnke, 1996) and other unpublished data, see details in (Dunne et al., 2007; Dunne et al., 2012). It is important to note that the observationally-based estimates of both the carbonate concentration and MAR have insufficient spatial resolution to adequately resolve continental margins (Fig. 2 top panel). The second map was generated using a metamodel derived from a process-based approach (including satellite-derived carbonate export

production, and water-chemistry-dependent dissolution in the water column and sediments) that was minimally-tuned using the sedimentary data (Dunne et al., 2012). The metamodel map shows similar patterns in the deep sea, but better resolves the bathymetric imprint on carbonate preservation (Fig. 2 bottom panel). The two approaches provide slightly different spatial



distributions of burial, allowing a test of the sensitivity of our reconstructions to this uncertainty (see Table 1). Although the absolute values differ, the relative changes in our reconstructed global carbonate burial over time are insensitive to the choice of maps (see appendices A).

The two modern maps attempt to accurately represent the spatial distribution of the modern $CaCO_3$ burial flux. Sediment cores, however, often target rapidly accumulating depocenters in order to provide greater temporal resolution for paleocenographic reconstructions. As a result, we expect a bias towards high accumulation rates, which is confirmed in Fig. 2. We address this sampling bias when evaluating uncertainties (see Appendices).

|  | Sed. Data | Metamodel |
|---|---|---|
| Atlantic | 41% | 48% |
| Pacific | 25% | 27% |
| Indian | 34% | 24% |
| Arctic | 1% | 1% |
| Antarctic | 0% | 0% |
| Sum (gCaCO₃/yr) | 1.14 10¹⁵ | 1.05 10¹⁵ |

**Table 1: Ocean basin contribution to the global modern CaCO₃ flux for the two approaches (see also Fig. 2, and**
**province map 1 in Fig. A1.**

### 3.3 Relative changes in burial rate from sediment core database

We created a database including surface- and downcore sediment datasets, retrieved from the NOAA (ftp://ftp.ncdc.noaa.gov/pub/data/paleo/paleocean/sediment_files/) and PANGAEA (http://www.pangaea.de) online repositories. All the datasets that contained $CaCO_3$ concentrations, $CaCO_3$ accumulation rates, age models and sediment
density values were taken into consideration (for a complete detailed description of the procedure the reader is referred to Cartapanis et al. (2016)). Whenever available, [230]Th-normalized sediment accumulation rates were used. Where these measurements were not available, sedimentation rates were determined at the depth of each carbonate measurement from the reported age-depth relationship. When possible, we used reported dry bulk density values. Otherwise, we assumed a constant value of 0.9 g*cm[-3], to convert linear accumulation rates to MAR. Each sedimentary record was expressed as a fraction of
the respective mean Holocene value.

The mean LGM:Holocene ratio was determined for each province, and multiplied by the corresponding modern burial rate to estimate the absolute value of carbonate burial during the LGM. Note that we used the geometric mean of available records in each province, given that the LGM:Holocene ratios of individual cores are log-normally distributed. Moreover, if in a single province we have two MAR records, one showing 2 times higher MAR during the glacial while the other shows only
half of modern during the glacial, the best guess would be that the true change in sedimentation was negligible. This is better



represented by the geometric mean of (2;0.5)=1 (no change) while the arithmetic mean (2;0.5)=1.25 would bias the final mean toward high values. The carbonate burial of all provinces is summed to provide the LGM global burial rate.

The same procedure was applied to calculate global carbonate sediment content and bulk MAR at every 1 kyr time step over the past 150 kyr. Any time interval within a sediment record for which no measurement was present over a period longer

than 10 kyr was excluded from the calculation of the regional stacks, leading to a total of 637 sediment time-series, from which 370 records were used to calculate the LGM:Holocene ratio (Fig. 3-4), while 170 to 130 records were used for periods older than 40 kyr. When no record was available for a specific province, and/or for a specific time period, the province flux was assumed to be constant and equal to modern values. The provinces for which no downcore record was available represent 0 to 18 % of the ocean surface, corresponding to 0 to 12 % of modern burial (this metric will be referred as

"coverage" in the following discussion) for the LGM (18-25 kyr, Fig. 3). As a consequence, the reduction of the coverage during older times tends diminishing the difference from the modern situation, a rather conservative assumption. The available records distributes over the entire depth range (100 above 1,000 m, 103 between 1,000 and 2,000 m, 137 records between 2,000 and 3,000 m, 290 between 3,000 and 4,000 m, and 106 records below 4,000 m deep).

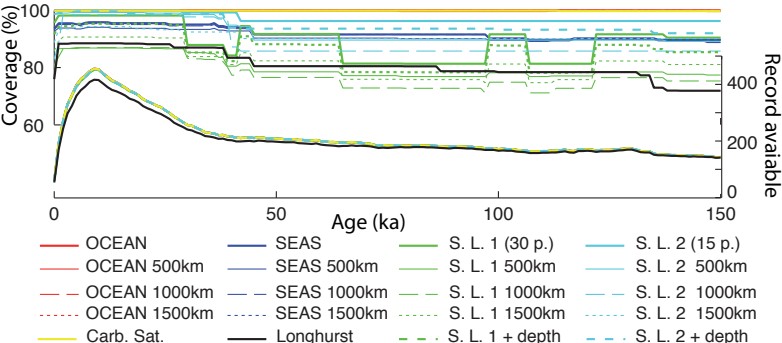

**Figure 3: Coverage and number of records used for each province-map. Coverage is calculated by summing the modern burial rates in each province for which at least one sedimentary record is available, divided by the modern burial flux. The modern map used corresponds to the top panel of Fig 2.**

### 3.5 Mass balance calculation

In order to estimate the impact of varying burial of carbonate in deep sea sediments during ice ages on the carbon and

alkalinity budgets, we calculate the running mass balances over the last full glacial cycle using the input and output fluxes for both carbon and alkalinity as illustrated in Figure 1. The deep sea $CO_3^{2-}$ concentration was calculated assuming a simple air-sea partitioning scheme of carbon, depending on the whole ocean alkalinity and assuming constant shallow/deep partitioning of the ocean carbon inventory (Sarmiento and Toggweiler, 1984).



The input flux to the ocean-atmosphere system consists of volcanic emissions and the erosion of continental surfaces (due to the dissolution of carbonate and oxidation of sedimentary organic carbon), while the output term corresponds to carbon burial in sediment, divided into four components: organic and inorganic, for each of the shelf and deep-sea environments. The input flux of alkalinity is given by the erosion of continental surfaces, including both carbonate and silicate weathering, while the flux out of the system is given by burial of carbonates in marine sediment.

It is important to note that we have not considered variable erosion fluxes in our reconstructions due to lack of reconstruction. Ref. (von Blanckenburg et al., 2015) have argued that erosion was, indeed, constant over glacial cycles, while others have argued that erosion was lower (Vance et al., 2009) or higher (Herman et al., 2013) during glacials. Erosion contributes both carbon and alkalinity, and although it remains difficult to evaluate the global impacts of changes in erosion, they may have been substantial.

Our simulations prescribe a range of possible scenarios for variable input and output fluxes, in order to estimate the resulting transient changes of carbon and alkalinity inventories and $\delta^{13}C$. The deep-sea output fluxes are derived from our sedimentary reconstructions, while the shallow output fluxes and the input flux are varied in order to illustrate hypothetical (yet realistic) scenarios. In total we include 8 different scenarios, to illustrate the impact of each variable in isolation, as well as in combination with others (Table 2). In all cases, we vary the fluxes over feasible ranges using a Monte Carlo approach (1,000 transient simulation per experiments).

A critical component of the simulations is the requirement for long-term balance over a full glacial cycle. The fact that $CO_2$ and $\delta^{13}C$ displays a very stationary mean value over several glacial cycles (Zeebe and Caldeira, 2008) suggests that the system is close to steady state over each full glacial cycle. Therefore, assuming that the system was at steady state over the last glacial cycle, we calculated the input fluxes of carbon and alkalinity that would have been necessary to balance the prescribed burial (as well as prescribed input changes, if any) over the full glacial cycle for a given scenario, and provided those as a constant input flux throughout the cycle. Thus, although the carbon and alkalinity fluxes are out of balance at each point within the cycle, they always integrate to zero net change between 0 and 123 kyr. Similarly, the isotopic composition of the system is prescribed by setting the $\delta^{13}C$ of the erosion flux at a value that balances the other fluxes, consistent with the long-term steady state assumption. Although forcing the carbon isotopes to be equal at start and end points does ignore a small long-term drift in whole-ocean $\delta^{13}C$ (Oliver et al., 2009), it provides a clearer illustration of the isotopic implications of inventory changes over a single glacial cycle. We note that this strategy results in different total input fluxes (and therefore different residence times of the active pool) for each experiment.

To account for uncertainties in the modern fluxes (Fig. 1), the relative changes in the prescribed time series are multiplied by an estimate of the global modern flux, which is generated using a distribution that accounts for uncertainty in modern global burial estimates (see Table 2, and appendices B). The forcing thus accounts for our reconstruction and their uncertainties, as well as the uncertainties in the modern burial estimates. The same procedure was applied for organic carbon burial variations in the deep ocean as shown in (Cartapanis et al., 2016), but using a different distribution (Table 2, see details figure A6).



| | exp. 1 | exp. 2 | exp. 3 | exp. 4 | exp. 5 | exp. 6 | exp. 7 | exp. 8 |
|---|---|---|---|---|---|---|---|---|
| | Deep carb | Deep toc | exp1+exp2 | Shelves | exp3+exp4 | Volc. | exp6+exp3 | exp5+exp6 |
| Shelves TOC | 0.15 cte | 0.15 cte | 0.15 cte | 0.15 | 0.15 | 0.15 cte | 0.15 cte | 0.15 |
| Shelves TOC (std) | 0 | 0 | 0 | gamma distr. 1 | gamma distr. 1 | 0 | 0 | gamma distr. 1 |
| Shelves CaCO$_3$ | 0.15 cte | 0.15 cte | 0.15 cte | 0.13 | 0.13 | 0.15 cte | 0.15 cte | 0.13 |
| Shelves CaCO$_3$(std) | 0 | 0 | 0 | 0.04 | 0.04 | 0 | 0 | 0.04 |
| Deep TOC | 0.07 cte | 0.07 | 0.07 | 0.07 cte | 0.07 | 0.16 cte | 0.07 | 0.07 |
| Deep TOC (std) | 0 | gamma distr. 2 | gamma distr. 2 | 0 | gamma distr. 2 | 0 | 0 | gamma distr. 2 |
| Deep CaCO$_3$ | 0.13 | 0.13 cte | 0.13 | 0.13 cte | 0.13 | 0.13 cte | 0.13 | 0.13 |
| Deep CaCO$_3$(std) | 0.03 | 0 | 0.03 | 0 | 0.03 | 0 | 0 | 0.03 |
| volc | cte | cte | cte | cte | cte | variable | variable | variable |

**Table 2: Modern burial distribution (PgC.kyr-1) used for the Monte Carlo simulations (mean and standard deviation). When the distribution type is not specified, a normal distribution was used, and "cte" indicates constant flux over times. Results of experiments 1 and 2, 3 to 5, and 6 to 8, are shown in figure 7, 8, and 10 respectively. The gamma distributions used for organic carbon burial are shown in Fig. A6.**

## 4. Results

### 4.1 Assessing changes in MAR and their role on the global estimates

Since carbonate MAR depends on both the sedimentation rate and sedimentary carbonate concentration, we first calculate the global bulk MAR by considering only relative changes in the sedimentation rates and a modern bulk MAR map (Jahnke, 1996). This reconstruction (Fig. 4, 5A, and Table 2) shows generally higher bulk sediment burial during peak glacials and deglaciations, broadly consistent with previous studies (Covault and Graham, 2010). However, the reconstructions display a pronounced increasing trend towards the present for all the province-map scenarios (Fig 5A). This is partly due to the absence of density data for many cores, which tends to underestimate burial for older, more compacted sediment. Moreover, the apparent sedimentation rate decreases as a power law function of the intervals between two age controls (Schumer and Jerolmack, 2009), suggesting that sedimentation rates are likely underestimated as age increases, when the age model accuracy is generally lower compared to more recent periods.





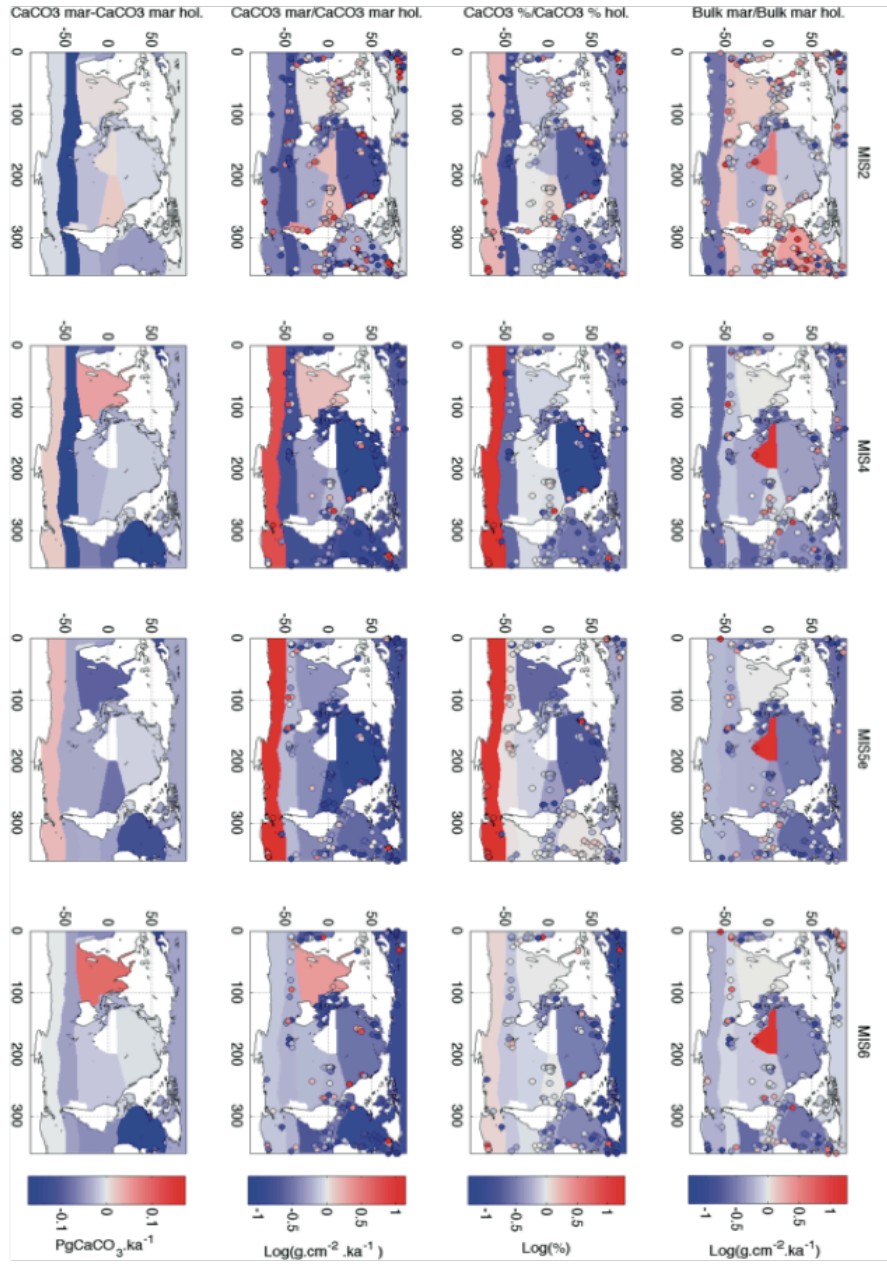

 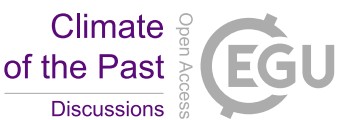

**Figure 4: Relative changes in bulk sediment MAR, carbonate content, and carbonate MAR, and absolute changes in carbonate MAR for province map S.L.2 and for LGM, MIS4, MIS5e and MIS6 (note that the absolute carbonate flux is not area normalized). The modern map used here corresponds to the top panel of Fig 2. Note that the data shown here are not corrected for the long-term trend of accumulation rate, as the correction is calculated and applied at the**

**global burial level.**

Because of the lack of density information for individual cores, we chose to correct for compaction at the global scale by assuming that the global bulk burial rate was equal during the late Holocene and MIS5e, and subtracting the intervening trend, an assumption that could be revisited in future work as more information becomes available. Given this assumption, we corrected the long-term trend of the global aggregate bulk sedimentation rate for each province-map using a least squares

spline (Fig. 5 A and B). The detrended $CaCO_3$ MAR reconstruction (Fig. 5F) is much more similar to the globally-flux-weighted $CaCO_3$ concentration record (Fig. 5D) and suggests that global bulk MAR in deep sea sediment peaked during glacial maxima and deglaciations, as well as during MIS4 (114 %, 104 % and 105 % relative to the Holocene MAR for the LGM, MIS4 and MIS5e respectively). The LGM to Holocene change is consistent with, but smaller than, a recent estimate (125 % +/- 15 %) that was made using a smaller but partially overlapping set of cores that included organic carbon

measurements (Cartapanis et al., 2016).

We performed a Principal Component Analysis (PCA) on the MAR time series in order to assess the overall spatial and temporal variability (Fig. 6). For this calculation we used only the records that cover at least 80 % of the studied interval. The first principal component of the bulk MAR shows variability that is similar to the bulk MAR reconstruction, and the sedimentation rate time series are broadly correlated with the sedimentation rate PC1, without showing any consistent

geographic pattern.

We then performed a similar analysis, using only the relative changes in the sedimentary carbonate content (assuming constant sedimentation rates), and the modern carbonate MAR map as a reference (Dunne et al., 2012). This analysis shows that the carbonate content in deep-sea sediment (weighted by modern carbonate MAR) was consistently lower during glacials compared to interglacials, with the lowest values reported during MIS4 and MIS2 (Fig. 4, Fig. 5D and 6). The first

principal component of the carbonate content records is similar to the global reconstruction, and the correlation map between carbonate content time series and carbonate content PC1 shows a clear anti-phasing between the eastern Pacific and the rest of the world ocean (Fig. 6). While carbonate content was lower during glacial in the Atlantic, it was substantially higher in the eastern Pacific.



| | CARB MAR corrected | | | | | | | | | | | |
|---|---|---|---|---|---|---|---|---|---|---|---|---|
| | MIS2 | | | MIS4 | | | MIS5 | | | MIS6 | | |
| experiment | # rec | cov. | MAR% | # rec | cov. | MAR% | # rec | cov. | MAR% | # rec | cov. | MAR% |
| Longhurst | 343 | 88 | 99 | 196 | 80 | 96 | 175 | 78 | 101 | 170 | 78 | 102 |
| OCEAN1000KM | 370 | 100 | 80 | 209 | 100 | 69 | 182 | 100 | 92 | 174 | 100 | 88 |
| OCEAN1500KM | 370 | 100 | 84 | 209 | 100 | 74 | 182 | 100 | 92 | 174 | 100 | 87 |
| OCEAN500KM | 370 | 100 | 72 | 209 | 100 | 64 | 182 | 100 | 90 | 174 | 100 | 84 |
| OCEAN | 370 | 100 | 78 | 209 | 100 | 65 | 182 | 100 | 90 | 174 | 100 | 83 |
| SEAS1000KM | 370 | 95 | 82 | 209 | 92 | 73 | 182 | 90 | 95 | 174 | 90 | 94 |
| SEAS1500KM | 370 | 95 | 84 | 209 | 92 | 79 | 182 | 90 | 96 | 174 | 90 | 95 |
| SEAS500KM | 370 | 94 | 75 | 209 | 90 | 70 | 182 | 90 | 94 | 174 | 90 | 94 |
| SEAS | 370 | 95 | 78 | 209 | 92 | 71 | 182 | 90 | 94 | 174 | 90 | 93 |
| S. L. 1_1000KM | 368 | 87 | 91 | 209 | 77 | 84 | 181 | 76 | 98 | 173 | 76 | 95 |
| S. L. 1_1500KM | 368 | 91 | 90 | 209 | 82 | 89 | 181 | 82 | 94 | 173 | 82 | 94 |
| S. L. 1_500KM | 368 | 87 | 87 | 209 | 79 | 80 | 181 | 78 | 99 | 173 | 78 | 95 |
| S. L. 1 | 368 | 98 | 93 | 209 | 92 | 85 | 181 | 92 | 98 | 173 | 92 | 98 |
| S. L. 1_depth | 368 | 95 | 91 | 209 | 88 | 81 | 181 | 88 | 98 | 173 | 88 | 97 |
| Carb Sat | 370 | 100 | 76 | 209 | 100 | 63 | 182 | 100 | 91 | 174 | 100 | 83 |
| S. L. 2_1000KM | 369 | 98 | 92 | 209 | 86 | 83 | 181 | 86 | 96 | 173 | 86 | 95 |
| S. L. 2_1500KM | 369 | 94 | 88 | 209 | 90 | 87 | 181 | 90 | 91 | 173 | 90 | 94 |
| S. L. 2_500KM | 369 | 99 | 84 | 209 | 90 | 80 | 181 | 90 | 94 | 173 | 90 | 96 |
| S. L. 2 | 369 | 100 | 87 | 209 | 96 | 89 | 181 | 96 | 91 | 173 | 96 | 96 |
| S. L. 2_depth | 369 | 99 | 87 | 209 | 94 | 82 | 181 | 93 | 92 | 173 | 93 | 97 |
| | | | | | | | | | | | | |
| Mean | 368 | 96 | 85 | 208 | 91 | 78 | 181 | 91 | 94 | 173 | 90 | 93 |
| Standard Dev. | 5.9 | 4.5 | 6.9 | 2.9 | 7.3 | 9.2 | 1.5 | 7.5 | 3.3 | 0.9 | 7.5 | 5.3 |

Table 3: Summary of estimated changes in carbonate MAR corrected from long-term trend for each province map. Number of records used, mean coverage (%), and mean MAR (in % of modern value), for four intervals (LGM, MIS4, MIS5e and MIS6), for each province-map. The results correspond to the calculations of temporal variability shown in Fig. 5 F. The modern map used for all calculations corresponds to the top panel of Fig. 2.



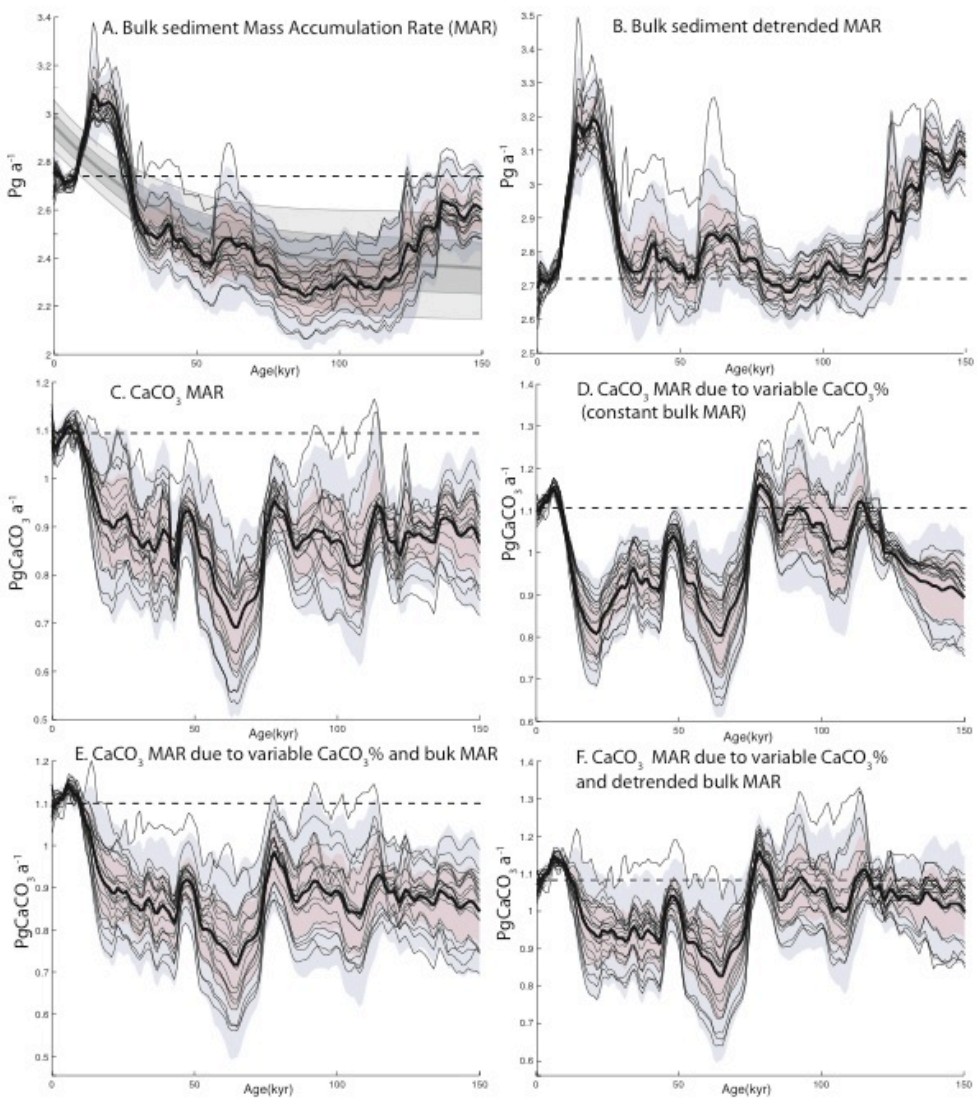

**Figure 5: CaCO₃ MAR timeseries, considering different aspects of variability. A. Bulk MAR time series distribution based on the 20 province-map scenarios (mean, black line; ±1σ, red area; ±2σ, blue area), and the distribution of the long term-trend correction for each province-map (mean, black line; ±1σ, bark grey; ±2σ, light grey) using a least-**



**square spline model. B. As in panel A, but corrected for long-term trend. C. Global CaCO₃ MAR reconstruction based on CaCO₃ uncorrected MAR timeseries. D. Global CaCO₃ MAR reconstruction calculated only from CaCO₃ content, assuming a constant bulk sedimentation rate. E. Global CaCO3 MAR reconstruction obtained by multiplying relative changes in the bulk MAR with CaCO₃ MAR based on CaCO₃ content time series only. F. As in**

**panel E, but with MAR corrected for long-term trend. On each panel, the thin black lines correspond to the 20 individual province-map scenarios. The modern map used for all calculations corresponds to the top panel of Fig 2.**

Finally, we calculated global carbonate MAR variations using 2 different methods. First we calculated global carbonate MARs using the relative changes in sedimentary carbonate MAR time series (Fig. 4, Fig. 5C). A simplified method consists in calculating the product of the reconstruction based on content, with the relative changes in bulk MAR reconstruction for

each scenario (Fig. 5E), in order to account for the impact of bulk MAR on the content.
As both methods provided very similar results, we finally calculated carbonate MAR as the product of the relative changes in corrected bulk sediment MAR with the reconstruction based on carbonate content only (Fig. 5F, see details in Fig. A4). This final reconstruction thus accounts for variations in the bulk MAR, as well as the variations in the content, but corrected for the long-term trend. This trend is directly inferred from the bulk MAR, which is the component of carbonate MAR

responsible for that trend (Fig. 5 A, and B, corresponding to bulk MAR and bulk MAR detrended and 5 D, corresponding to the carbonate MAR due to carbonate content).





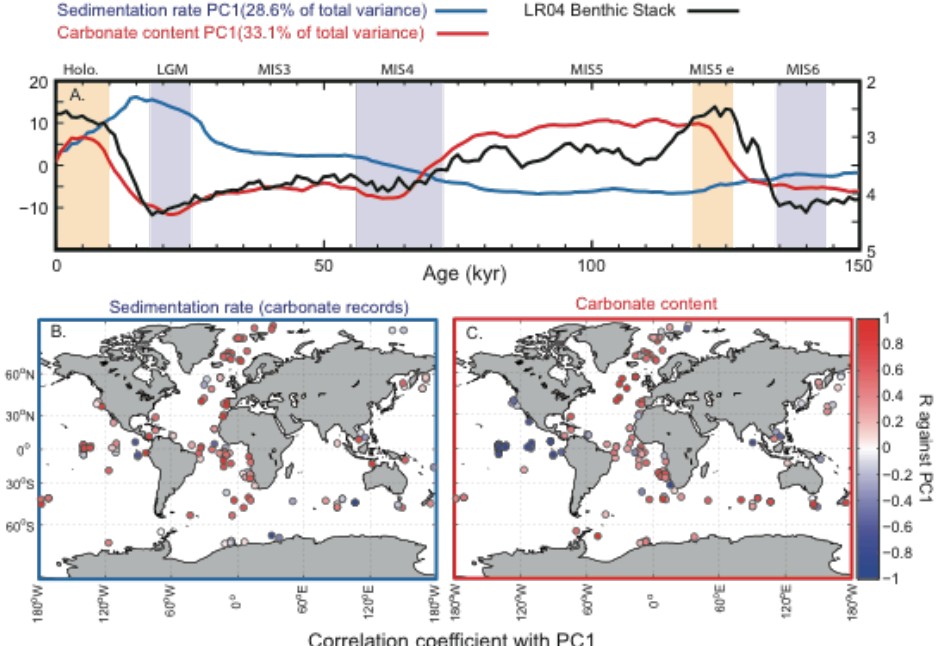

Correlation coefficient with PC1

**Figure 6: Principal Component Analyses (PCA) on sedimentation rates and carbonate content variations. A PCA was performed on the bulk MAR time series, as well as for the carbonate content time series. In this analysis, we used only records that cover 80 % of the studied interval. The first principal component (PC1) is displayed in the top panel for both sedimentation rates and carbonates content and explains 28.6 % and 33.1 % of the total variance respectively. The bottom panels show the correlation coefficients for individual records of sedimentation rates and carbonate content, against their respective PC1.**

As shown in Fig. 5 F, the reconstruction indicates that global mean deep-sea $CaCO_3$ burial was not significantly different from present-day throughout Marine Isotopic Stage 5 (MIS 5), dropped to the lowest rates during MIS 4, and then remained below interglacial rates until the deglaciation (see also details in Appendices).

## 4.2 Deep sea burial impact on carbon and alkalinity inventories

Assuming a constant alkalinity flux to the ocean, and all other factors being equal, our reconstruction of carbonate burial in the deep ocean coupled with uncertainties in modern fluxes (Table2) implies changes in the alkalinity inventory that could





reach up to 100 μequ/kg, with a minimum centered around ~70 kyr near the MIS4/MIS5 transition, gradually rising to a maximum during the early deglaciation at 15 kyr (Fig. 7). This general trend is a very robust consequence of our reconstructed carbonate burial, with a most probable amplitude of 70 +/-30 μequ/kg.

Changes in organic carbon burial in the deep sea followed a markedly different progression, with larger (rather than smaller)

5   burial during peak glacial and deglaciation (Cartapanis et al., 2016), causing the C inventory to rise over MIS 5 by 400 (150 to 950) PgC, equivalent to 20 (10 to 50) μmol/kg in the global ocean (Fig. 7 B), and to decrease by the same amount from MIS3 to the early Holocene. Summing the two deep sea burial fluxes together leads to a slightly muted change in the carbon inventory, as compared to alkalinity. The changes in deep sea organic carbon burial would have decreased the isotopic composition of the system carbon pool during the early glacial, by 0.2 (0.1 to 0.5) ‰ during MIS4 (Fig. 8 A). It is important

10   to note that for these 3 first experiments, mean C in put flux required to compensate for the outputs is close to 500 PgC/kyr, which exceeds the standard estimates for present-day carbon input.

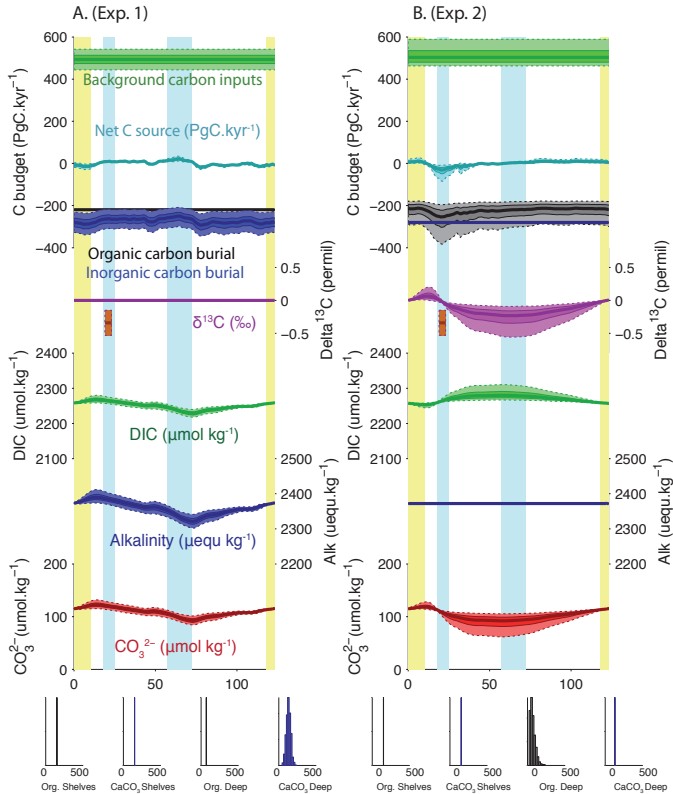





**Figure 7: Simulated impacts of reconstructed geological fluxes on the global carbon cycle. A. Deep sea CaCO₃ burial only. B. Deep sea organic carbon burial only. For each panel, the uppermost curves show the carbon budget terms (constant background carbon input, organic, inorganic carbon burial, and sum) used to force the simulations. For each time series, the mean of the scenarios (thick line), ±1σ, and ±2σ are shown. Below are shown the simulated δ¹³C**

5  **of the total active carbon pool, deep ocean alkalinity, deep ocean DIC, deep ocean CO₃²⁻ (µmol/kg). An estimate for the whole ocean δ¹³C by Peterson et al. (2014) during the last glacial maximum is also shown in orange (with upper and lower estimate). The distribution of the modern fluxes used to drive the scenarios is shown by the histograms at the bottom of each panel.**

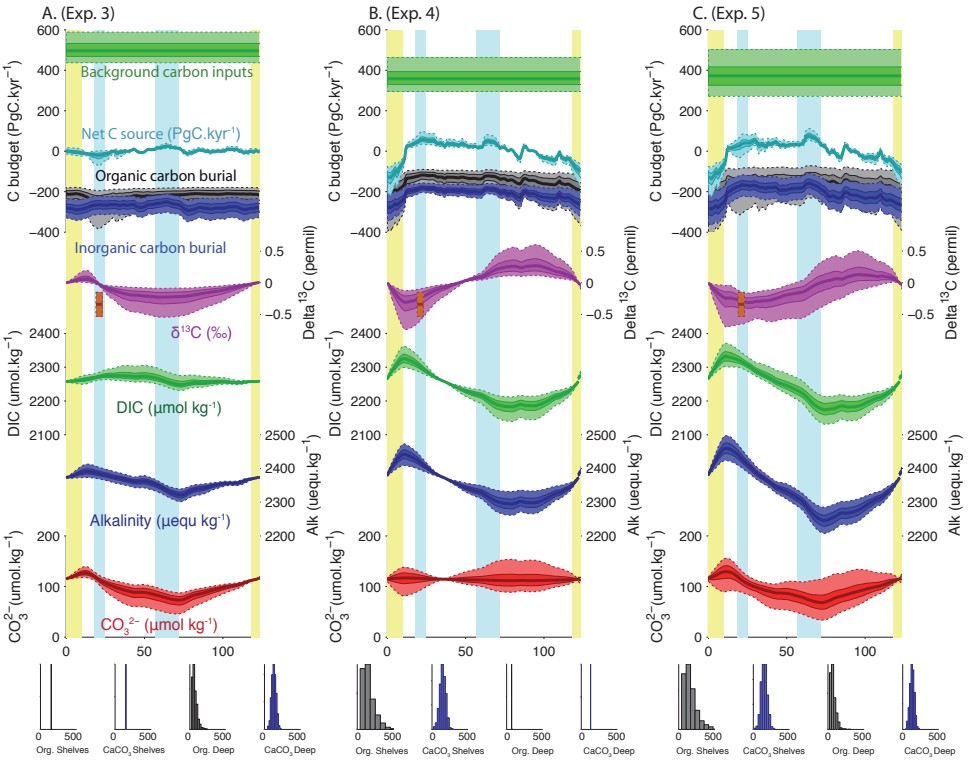



**Figure 8: Simulated impacts of reconstructed geological fluxes on the global carbon cycle. A. Deep sea burial (carbonate plus organic carbon). B. Shelf burial only. C. Total marine burial (deep + shelf). All quantities plotted as in Fig.7**

### 4.3 Shelf burial impact on carbon and alkalinity inventories

The burial of carbonate and organic carbon on shelves depends both on their production rate and their preservation. Preservation is particularly difficult to estimate, given the importance of sediment transport and redeposition in active coastal environments. Here, we use the very simple illustrative assumption that, globally, the burial over shelves was linearly proportional to the changes in the surface area available for sediment production and accumulation. Although this is a gross simplification of reality, it allows a first-order estimate of the potential magnitude of shelf burial changes on the global

carbon cycle (Wallmann et al., 2016).

We calculated the relative changes in the surface area bounded by the 0 and 100 m isobaths as well as the 0 and 200 m isobaths over time, using ETOPO01 (Amante and Eakins, 2009) and a global sea level reconstruction curve (Grant et al., 2012). Because of the distribution of the global hypsometric curve (Fig. 9 A), the surface area of the 0-100 m depth range was reduced by up to 60 % during the last glacial, which is certain to have led to reduced burial (Fig. 9 B), consistent with

(Vecsei and Berger, 2004).

Our results show that organic and inorganic MAR variations driven by changes in sea level could have increased DIC and alkalinity inventories by up to 150 μmol/kg and 150 μequ/kg from MIS5 to the last deglaciation, because of reduced burial during glacial times (Fig. 9 B). Because organic carbon burial is strongly focused in shallow environments, reduced shelf burial during glacials would have reduced $\delta^{13}C$ of the active carbon pool by up to 1 ‰ from MIS5 to the early deglaciation.

The combined effect of burial in deep and shelf seas tends to increase DIC and ALK inventories during glacials, relative to interglacials, by 150 (50 to 250) μmol/kg and 150 (100 to 300) μequ/kg respectively. The large uncertainties in modern burial fluxes result in large uncertainties in the ratio of DIC:ALK changes, with important implications.



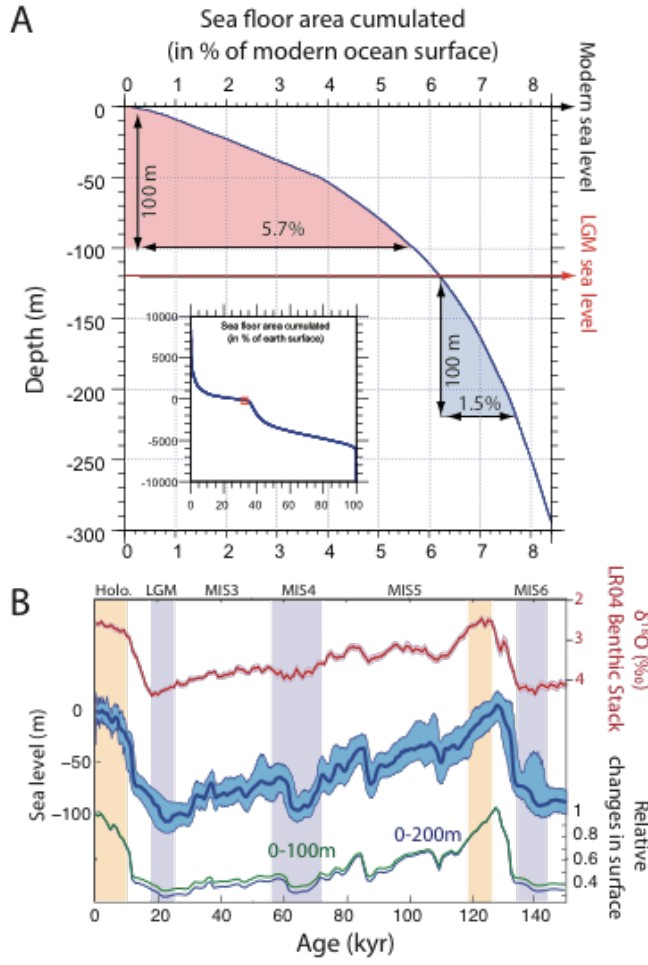

**Figure 9: A. Global hypsometric curve based on ETOPO01 (Amante and Eakins, 2009); B. LR04 benthic stack (Lisiecki and Raymo, 2005); sea level reconstruction (Grant et al., 2012); and calculated changes in the surface of the 0-100m and 0-200m depth interval relatively to modern situation.**



### 4.4. Combined impact of sedimentary burial and carbon input variations

It has been suggested that the rate of carbon input to the active pool increased during glacial maxima (Lund et al., 2016; Ronge et al., 2016) and/or deglaciations (Huybers and Langmuir, 2009; Stott et al., 2009; Zech, 2012). The magnitude of these carbon input variations is highly uncertain, and reconstructions of relative changes in volcanic outgassing rates over time suggest that the volcanic carbon input was notably delayed, relative to the deglacial increase in $pCO_2$ recorded in ice cores, requiring other mechanisms to be important at this time Huybers and Langmuir (2009) and Roth and Joos (2012). Nonetheless, it would appear likely that changes in geological carbon inputs over the glacial cycle were significant. We do not attempt to provide an exhaustive test of the existing hypotheses for temporally-variable carbon input, which includes peat and permafrost changes as well as volcanic outgassing, but simply use a pulse of carbon input during the deglaciation, which follows the temporal changes given by the reconstruction of changes in terrestrial volcanic emission (Huybers and Langmuir, 2009) to produce an illustrative variable carbon input to the active pool. The results can generally be extended to any kind of carbon inputs, though the isotopic impacts would differ.

The simulation with only variable carbon input produces similar results than to those estimated by Huybers and Langmuir (2009). When viewed over the glacial cycle, and given our requirement that the carbon inventory be the same at the beginning and end of the cycle, the main effect of a temporally-variable carbon input is to lower the background input rate. As a result, the C inventory gradually declines from interglacial to glacial maximum, and is then rapidly increased by the volcanic pulse (Fig. 10A).

When our reconstructed deep burial variations are also included, the magnitude of the MIS 5-4 carbonate ion peak is reduced relative to the deep-burial-only case (Fig. 10B). When shelf burial changes are also included, the mean carbonate ion variations are altered little, but the $\delta^{13}C$ changes significantly. In this case, the $\delta^{13}C$ of the active pool during the LGM is similar to that observed, even with no change in the terrestrial biosphere (Fig. 10C).

Notably, only the experiments including variable shelf burial (Fig. 8B and C and 10C) simulate Holocene input fluxes that overlap with the observational estimates summarized in Figure 1.



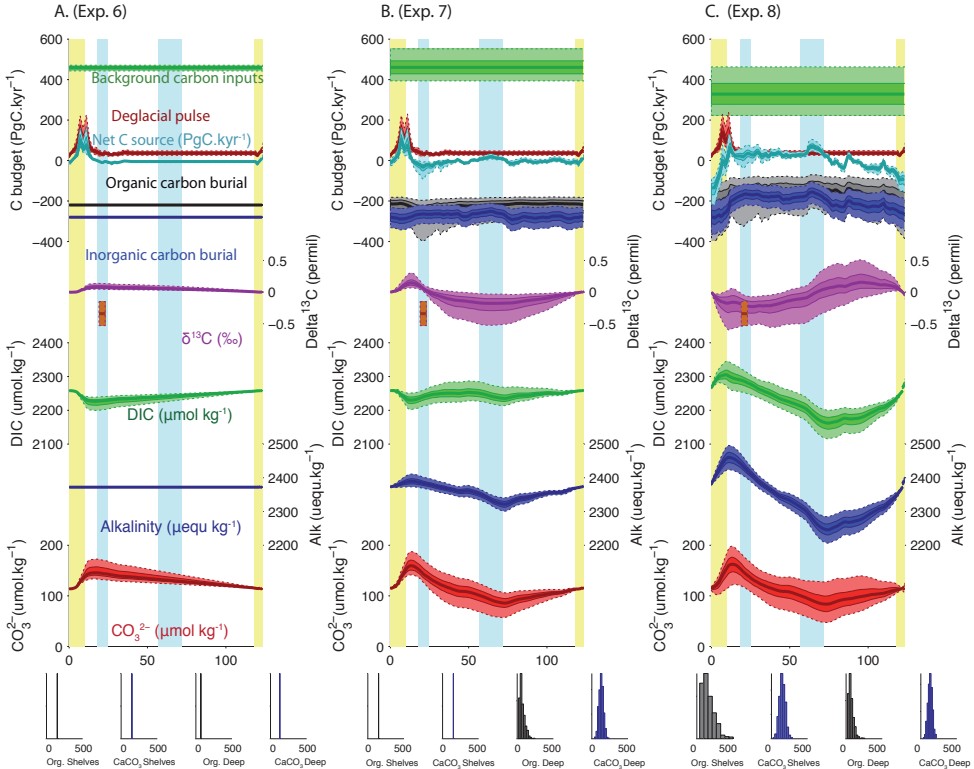

**Figure 10: Simulated impacts of reconstructed geological fluxes on the global carbon cycle. A. Deglacial carbon input pulses only; B. Deep sea burial plus deglacial carbon input pulses; C. All variables (Deep sea burial, shelves burial and deglacial carbon input pulses). All quantities plotted as in Figure 7, with the addition of time-varying carbon input pulses based on subaerial volcanic emission (Huybers and Langmuir, 2009) in red.**

## 5. Discussion

### 5.1 Carbonate burial in the deep ocean

We calculated individual time series of CaCO₃ mass accumulation rate (MAR) for hundreds of sediment cores, using previously-reported sedimentary CaCO₃ content, sedimentation rates and dry bulk density when available (SI). Following the method of Cartapanis et al. (2016), we used these individual records to calculate, at each 1 kyr time step over the past 150





kyr, the CaCO$_3$ MAR in oceanographically-defined provinces as a fraction of their Holocene values. Notably, these sediment cores record the actual, realized burial, not just a proxy thereof, although the preserved record may under-represent transient deposition and dissolution events, and the available sediment cores are undoubtedly subject to multiple biases (Appendix A). However, our broad spatial reconstruction would tend to minimize locally obfuscating factors, providing a robust

reconstruction of carbonate burial, globally. The modern flux in each province (Fig. 1) was then multiplied by the corresponding time-series of relative change, to obtain absolute changes in burial by province, and the uncertainty in the global burial flux was estimated using a bootstrap technique (Appendix A). The resulting reconstruction is sure to contain biases, and will hopefully be improved in future, however it was not obvious to us how the reconstruction could be improved significantly without a major expansion of the very large existing dataset.

As shown in Fig. 11, the reconstruction indicates that global mean deep-sea CaCO$_3$ burial was not significantly different from present-day throughout Marine Isotopic Stage 5 (MIS 5), dropped to the lowest rates during MIS 4, and then remained below interglacial rates until the deglaciation. The spatial pattern of change, which displays the well-known deep Pacific-Atlantic antiphasing ((Farrell and Prell, 1989; Hain et al., 2010), Fig. 6), is consistent with a previously reported shoaling of well-ventilated North Atlantic Deep Water (NADW) (Lynch-Stieglitz et al., 2007; Hain et al., 2010; Guihou et al., 2011) and

an enhanced soft tissue pump at the start of MIS 4. The associated increase of DIC-rich Antarctic Bottom Waters (AABW) would have decreased CO$_3^{2-}$ and favoured the dissolution of CaCO$_3$ in deep Atlantic sediments. An increased rain of organic matter to the deep sea, or a change in the rain ratio, such as may have resulted from a postulated shift toward proportionally more siliceous phytoplankton (Brzezinski et al., 2002) or a mid-latitude temperature decrease (Dunne et al., 2012), could have accentuated the DIC increase (Boyle, 1988; Hain et al., 2010; Yu et al., 2014; Galbraith and Jaccard, 2015).

The reduced flux of alkalinity removal in the Atlantic during the glacial would have increased the total alkalinity inventory of the ocean, which would have been expected to increase CO$_3^{2-}$ and consequently carbonate preservation elsewhere, a compensation long understood to have occurred in the Pacific Ocean (Farrell and Prell, 1989). However, our calculations suggest that reduced burial in the Atlantic Ocean wasn't entirely compensated in the Pacific Ocean. This implies that, if the alkalinity inputs from weathering had remained constant (as assumed in our mass balance calculations), excess alkalinity in

the global ocean must have built up during glacials as compared to interglacials. However, this apparently did not lead to significantly higher rates of carbonate burial. Additionally, despite notable pulses of CaCO$_3$ burial in the deep ocean in some locations during the deglaciation (Jaccard et al., 2005; Dubois et al., 2010; Rickaby et al., 2010; Jaccard et al., 2013), and in contrast with expectations (Broecker and Peng, 1987; Boyle, 1988; Sigman and Haug, 2003), our reconstructions suggest that preserved deep sea CaCO$_3$ burial itself did not drastically peak during the last deglaciation, although post depositional

dissolution cannot be excluded.



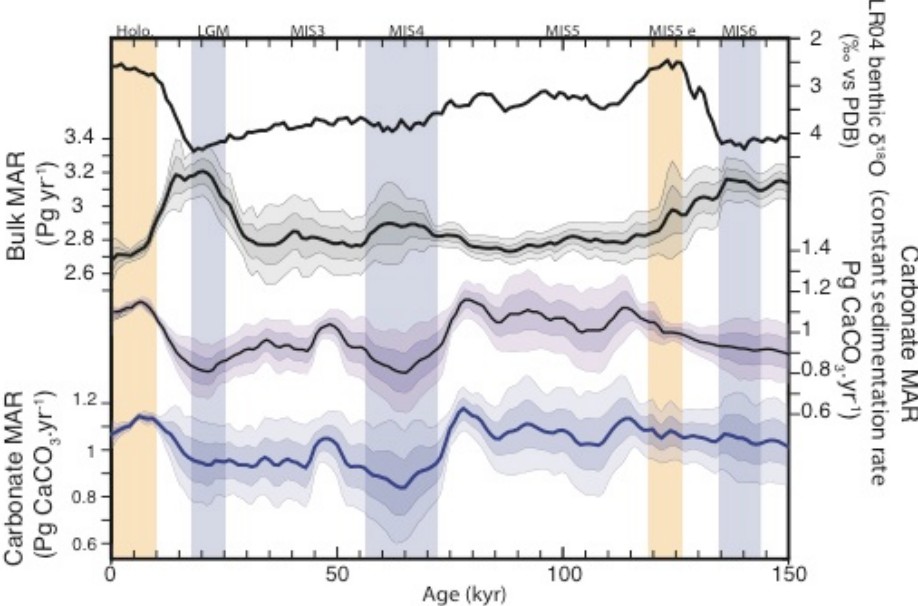

**Figure 11: Global changes in deep sea carbonate burial over the last glacial cycle. Curves show, from top to bottom, a global stack of benthic foraminifera δ $^{18}$O (Lisiecki and Raymo, 2005) reconstructed bulk sediment mass accumulation rate (MAR); CaCO₃ MAR assuming constant sedimentation rate; CaCO₃ MAR using reconstructed**

**sedimentation rate. The latter three curves show the mean, ±1σ, and ±2σ, see details in SI). Vertical bars highlight cold periods (MIS 2/LGM, MIS 4, and MIS 6) and full interglacial conditions (Holocene and MIS 5e).**

## 5.2 Changes in deep sea CO₃²⁻

Moreover, the fact that deep-sea CaCO₃ burial did not increase during the glacial, despite the loss of marginal carbonate depocenters due to sea level fall, may seem even more counterintuitive. If the alkalinity input flux from weathering had

10 remained constant (Gibbs and Kump, 1994), the loss of the coastal alkalinity sink should have led to a buildup of ocean alkalinity and a compensatory increase in deep-sea CaCO₃ burial (Opdyke and Walker, 1992). However, this chain of expectations does not consider possible increases in the DIC of the deep sea that could have counterbalanced the impact of an alkalinity increase on the $CO_3^{2-}$ there. Because the $CO_3^{2-}$ is dependent not only on ALK, but also on DIC (since $CO_3^{2-}$ = ALK – DIC), large changes in DIC will cause any compensation to deviate from constant ALK. In fact, it seems likely that

deep sea DIC changes would have arisen, both from enhanced carbon sequestration by saturation, soft tissue and



disequilibrium carbon storage (Eggleston and Galbraith, 2017; Ödalen et al., 2018), and changes in the active carbon/alkalinity inventory (Yu et al., 2013).

Our mass balance calculations resolve only changes in geological inputs and outputs, and do not include all additional changes in deep sea carbon storage. As a result, they can infer changes in $CO_3^2$ that are unrealistically large, and disagree with the relatively small changes in $CO_3^2$ that have been reconstructed from foraminiferal proxies (Yu et al., 2013; Yu et al., 2014). Changes in soft tissue and disequilibrium DIC in the deep sea, simulated in complex Earth system models driven with glacial boundary conditions, are of the same order as the $CO_3^{2-}$ excursions (order 40 uM each, (Eggleston and Galbraith, 2017)). We therefore suggest that the large simulated excursions of $CO_3^{2-}$ in Figures 8 and 10 reflect the incompleteness of the scenarios, largely due to the unresolved changes in deep sea soft tissue and disequilibrium DIC.

### 5.3. Important uncertainties in shelf burial

As shown in Figure 1, modern estimates suggest that approximately half of the global $CaCO_3$ burial, and most of the $C_{org}$ burial, occurs on continental margins. Unfortunately we do not have direct records of changes in shelf burial over time, so have instead evaluated the likely impact of global shelf burial changes by considering a simple but illustrative scenario, assuming linear relation between shelve surface variation and burial (Fig. 9).

However, organic and inorganic carbon burials have opposing effects on $CO_3^{2-}$, with the buildup of DIC during low sea level counteracting the buildup of alkalinity, so that the net effect depends on the $CaCO_3/C_{org}$ ratio of net burial. For a $CaCO_3/C_{org}$ ratio of approximately 1.3 (mol/mol), the opposing effects of DIC and alkalinity perfectly compensate, leading to muted changes $CO_3^{2-}$, and atmospheric $CO_2$, even with large burial and inventory changes. This sensitivity to the shelf burial ratio is illustrated in Figure 12. Given that estimates of the shelf burial ratio range widely and may have changed significantly over time, it does not appear possible to definitively conclude whether the net effect was to raise or lower $CO_2$ during glacial periods without improved observational constraints.



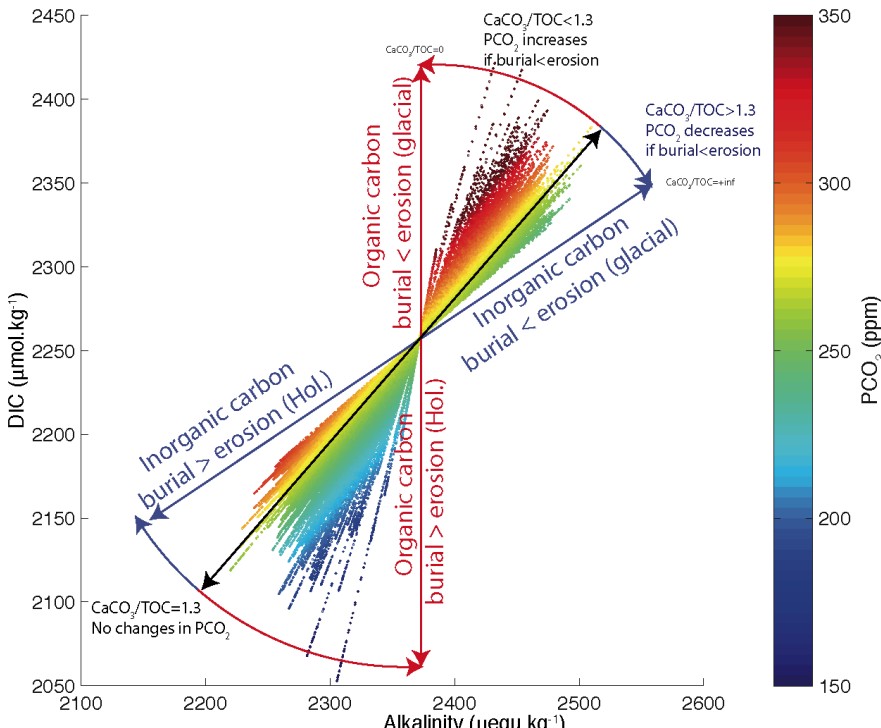

**Figure 12: Atmospheric PCO₂, DIC and alkalinity in the deep ocean for the shelf scenarios (Fig. 8 B). Each experiment appears as a straight line of dots and each dot corresponds to a 1 kyr timestep. The tilt of the individual scenario lines depends on the CaCO₃/C_org. burial ratio in shelf sediments. Blue and red lines correspond to 100 % carbonate and 100 % C_org. burial respectively.**





### 5.4 Implications of geological fluxes for whole ocean d13C and terrestrial biosphere

However, even if climatic impact of burial changes on shelves were muted, it would have left a significant signature in active carbon inventory $\delta^{13}C$, with a strong likelihood for lower $\delta^{13}C$ during MIS 3-2. This isotopic effect arises from the fact that $C_{org}$ burial is strongly focused on continental margins, compared to carbonate burial, which is more evenly distributed

between margins and the deep sea. Higher burial in the deep sea (Cartapanis et al., 2016) would be insufficient to compensate for reduced burial on shelves. The smaller proportion of low- $\delta^{13}C_{org}$ buried during periods of low sea level would have produced a downward drift of the $\delta^{13}C$ of the active carbon inventory (Wallmann et al., 2016) while enhanced volcanic activity could have raised $\delta^{13}C$ during deglaciation (Huybers and Langmuir, 2009; Roth and Joos, 2012).

This result is important, given the frequently-applied technique of calculating changes in the terrestrial biosphere using

reconstructed changes in whole-ocean $\delta^{13}C$ and assuming a constant inventory (Bird et al., 1996; Ciais et al., 2012; Menviel et al., 2012; Peterson et al., 2014; Schmittner and Somes, 2016). The magnitude of ocean $\delta^{13}C$ changes that result from changes in shelf burial can exceed the observed glacial-interglacial change of 0.32. Unfortunately, this invalidates the use of $\delta^{13}C$ as a direct monitor of changes in the terrestrial biosphere, and requires that new types of constraints on changes in terrestrial biosphere be developed.

In general, the mass balance calculations suggest that the terrestrial biosphere during the LGM was significantly larger than implied by the constant-inventory assumption. We note that using the highest modern $C_{org}$ burial flux (on the order of 500 PgC ky$^{-1}$) would imply a change of the $\delta^{13}C$ of the system on the order of 1 ‰ (Fig. 8 and 10), which is higher than reconstructed variability for $\delta^{13}C$ in the deep ocean, in which most of the carbon resides (Oliver et al., 2009; Peterson et al., 2014). This suggests that either highest estimates for modern burial over shelves (above 500 PgC ky$^{-1}$) are inconsistent with

the assumption of a 60 % decrease of the shelf burial during peak glacial, or that the terrestrial biosphere actually contained more biomass during the glacial than today, opposite the standard assumption. Although this is highly uncertain, such a change could have occurred through an expansion of peatlands and permafrost carbon during the glacial (Ciais et al., 2012). An expansion of terrestrial carbon storage in a colder state would perhaps be more consistent with the general expectation for carbon release from peatland and permafrost under anthropogenic warming (Ipcc, 2014), though it should be viewed as a

highly speculative possibility.

### 5.5 Non-steady-state Holocene carbon cycle

Our results show that the removal rate of carbon from the active pool was higher during the Holocene (prior to anthropogenic carbon release) compared to glacial. This result, which emerges from the fact that both shelf burial and deep sea carbonate burial were reduced during the glacial, indicates that the active carbon inventory was shrinking prior to

anthropogenic inputs. The estimates suggest that this imbalance was mostly attributable to the shelf burial, though at a rate that was very slow compared to subsequent fossil fuel emissions (~ 100-200 PgC kyr$^{-1}$ net loss vs. present-day emissions of ~ 10,000 PgC kyr$^{-1}$).





In support of this large glacial-interglacial variation in carbon burial rates, we find that only mass-balance calculations that include reduced burial during glacial infer background inputs of carbon that are consistent with the <400 PgC kyr$^{-1}$ geological inputs estimated in the literature (Fig. 1). Essentially, without slower burial during the glacial, modern-day burial estimates would require larger carbon input fluxes than supported by observational estimates in order to balance the long-term inventory.

## 6. Conclusions

We have made use of an extensive, quality-controlled database for marine sedimentary proxies, to present the first global quantitative reconstruction of carbon and alkalinity burial in deep-sea sediments over the last glacial cycle. Our reconstructions of carbonate and organic carbon burial in the deep ocean provide the first dynamic estimate of these major

components of carbon removal over a full glacial cycle.

Our reconstruction indicates that $CaCO_3$ removal fluxes in deep-sea sediment remained indistinguishable from the Holocene through MIS5, dropped to their lowest value during MIS4 (78 % ± 9 % of Holocene value), and remained lower than the Holocene during MIS2 (85 % ±7 %). The reduction of carbonate burial in the Atlantic Ocean during glacial due to enhanced soft tissue pump wasn't entirely compensated by increased burial in the Pacific Ocean implying the build up of alkalinity in

the glacial ocean, had input fluxes remained constant. We suggest that this weak compensation of burial was related to the parallel accumulation of DIC in the deep sea, which prevented high $CO_3^{2=}$. Moreover, the reduction in shelf burial during sea level lowstands, that is required for mass balance regarding to geological inputs, would have unquestionably grown the active carbon and alkalinity inventories, followed by reduction of the inventories when sea level rose and rapid burial resumed. The consequent changes in the global ratio of organic carbon:carbonate burial would have produced significant

changes in the d13C of the active carbon pool, raising important uncertainties in the use of whole ocean d13C to reconstruct changes in the terrestrial biosphere.

The emergent picture of these results suggests that geological carbon fluxes are more dynamic than frequently assumed, and that this includes not just the input fluxes (Huybers etc.), but also the output fluxes through marine burial. As a result, they need to be considered as an integral part of glacial-interglacial cycles.

**7. Acknowledgements**

O.C. and S.L.J. were funded by the Swiss National Science Foundation (grant PP00P2-144811). O.C. and D.B. were funded by the Canadian Institute for Advanced Research (CIFAR). E.D.G. acknowledges financial support from the Spanish Ministry of Economy and Competitiveness, through the María de Maeztu Programme for Centres/Units of Excellence in R&D (MDM-2015-0552).





**8. Appendices**

**A: Uncertainties in global bulk sediment and carbonate burial in the deep ocean over the past 150 kyr**

Our reconstructions are affected by multiple sources of uncertainty. We identify four different types of uncertainty in section A, and indicate whether each error is expected to be independent of the others. We then estimate the contributions from the

most important sources of uncertainty using bootstrap analyses (see section B).

**A.1 Uncertainty in the sedimentary records**

First, there is some uncertainty associated with the quantification of past changes in the $CaCO_3$ accumulation rate in each downcore record. This can be further subdivided into three uncertainties:

**a:** Measurement of $CaCO_3$ concentration. Analytical error order 3%. Independent.

**b:** Linear sediment accumulation rate. Potentially large over periods less than age control point spacing, but small (a few percent) when integrated over the span of multiple age control points. Independent.

**c:** Sediment density. Based on the literature, and using our database, we attempted to understand the relationships between depth in the sediment, sediment composition and sediment density. The density of the sediment is partly dependent on the carbonate content, but carbonate content influence on the density depends also on the density of the other components of the

sediment (Reghellin et al., 2013), so that it was not possible to accurately infer density variations from the changes in the carbonate content alone. Attempts to estimate density from other sedimentary variables based on existing calibrations (Dadey et al., 1992; Weber et al., 1997; Fortin et al., 2013) were not successful. Unassessed changes of sediment density are therefore likely to introduce uncertainty for many records, although the largest and most systematic change is likely to be the long-term increasing trend caused by compaction, which is addressed by the long-term trend correction. This source of

uncertainties is mostly independent, though it could possibly correlates with $CaCO_3$ content.

We assume that these errors are small (probably less than 10% relative error) and un-correlated with the other, larger sources of error given below, and therefore do not formally assess them.

**A.2 Representativity of local regions by sedimentary records**

Each downcore record may be biased relative to its local. Many paleoceanographic cores are purposefully collected from

rapidly-accumulating sites, in order to maximize temporal resolution. This intentional collection bias towards high sedimentation rate sites could affect the representativity of:

**a:** $CaCO_3$ concentration, due to differential transport and/or preservation of $CaCO_3$. Magnitude unknown, though the fact that paleoceanographers very frequently use high accumulation rate sites in order to infer regional changes in sediment composition implies that it is generally thought to be minor. Possibly correlated with 2b, through differential transport of

$CaCO_3$ vs. other sediment fractions.

**b:** Mass accumulation rate, if the accumulation rate in the cored location varies in a way that is not proportional to variation





of sedimentation rates in the local region. Magnitude unknown, but presumably large for some very rapidly-accumulating 'drift' sites where sediment focusing changes over time. These changes will cancel each other out to some degree if multiple cores in the same region are considered, since a relative decrease in sedimentation rate at one site must be compensated by a relative increase elsewhere, although there is no guarantee that the available sampling density is sufficient to achieve this.

5    Possibly correlated with 2a.

The uncertain representativity at the local scale (including both 2a and 2b) is addressed by bootstrap resampling of the available sedimentary records (see A4 and fig. A2).

### A.3 Province-maps

In order to account for spatial coherence in the geographic and bathymetric variability in global carbonate burial, we subdivided the world ocean into contiguous regions we refer to 'provinces'. As any subdivision will introduce inherent bias, we used a total of 20 different province-maps, allowing us to analyze the dependency of our results on the province-map used (Fig. A1). The first two maps simply subdivide the world ocean into major ocean basins (map 1), as well as a finer subdivision into seas based on the International Hydrographic Organization (1953) definition (map 3). Because carbonate export from the ocean surface largely depends on the biogeochemical conditions at the surface, we also used the annual climatology of the ocean biogeochemical provinces based on (Longhurst, 1995), recently updated by Reygondeau et al. (2013) (map 4). This map distinguishes 56 coherent provinces from a biogeochemical perspective. We simplified and adapted this map to match our sample distribution and came up with 2 sets of maps subdivided in 15 and 30 provinces, respectively (map 5 and 7). Water depth is also important for carbonate, given that carbonate solubility varies as a direct result of pressure. In order to resolve the possible influence of depth on our reconstruction, we separated the two modified Longhurst maps into a shallow and deep component, following the 1,500 m isobaths (map 6 and 8). Furthermore, given the potential importance of changes in carbonate preservation due to changes of water column chemistry, we subdivided the ocean according to the carbonate saturation state at ocean bottom ($\Omega_{cal}<0.8$; $0.8< \Omega_{cal} <1.2$; $\Omega_{cal}>1.2$) (Dunne et al., 2012) (map 2). Finally, in order to account for the potential heterogeneity close to coasts, we subdivided the ocean map, the sea map, and the two simplified Longhurst maps into coastal and open ocean provinces, using three different thresholds of distance from the closest shoreline: 500 km, 1000 km and 1500 km (maps 9 to 20; see details of the results for individual provinces maps in figure A2, A4 and A5).



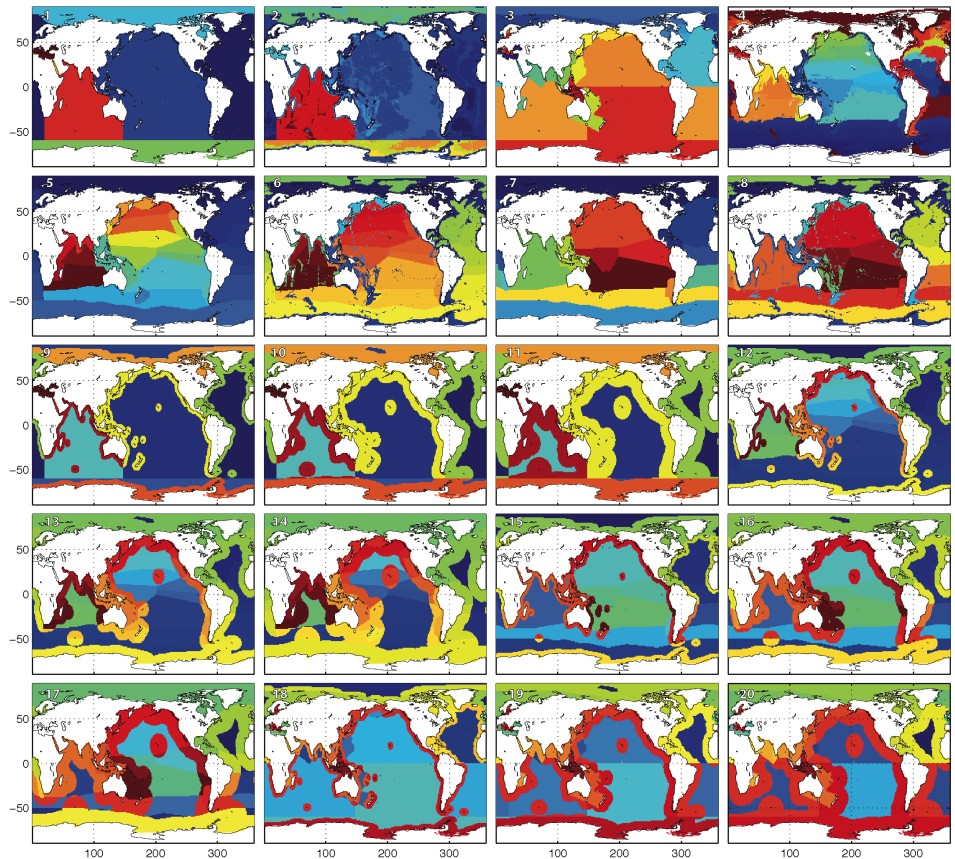

**Figure A1: Province maps used in this study, see details in text. 1. Ocean; 2. Carbonate saturation; 3. Seas; 4. Longhurst; 5. Simplified Lonhurst 1 (S.L.1). 6. S.L.1+depth; 7. S.L.2; 8. SL2+depth; 9. Ocean 500 km; 10. Ocean 1000 km; 11. Ocean 1500 km; 12. S.L.1 500 km; 13. S.L.1 1000 km; 14. S.L.1 1500 km; 15. S.L.2 500 km; 16. S.L.2 1000 km; 17. S.L.2 1500 km; 18. Seas 500 km; 19. Seas 1000 km; 20. Seas 1500 km.**

**A.4 Heterogeneity within provinces**

As discussed above, sediment cores represent local regions within provinces. Temporal trends in the burial flux of CaCO$_3$ within any given local region will differ, to some degree, from the trend of the larger province in which the region is located. The larger the province, the more severe this problem is likely to be. This heterogeneity is alleviated, to some degree, by



averaging multiple records within a given province (Fig A1).

It is important to consider that there may be a systematic bias, given that cores tend to be located in the most rapidly-accumulating regions of provinces, including continental slopes. However, although this would lead to incorrect inferences regarding changes in $CaCO_3$ MAR in the slowly-accumulating regions of the province, it is less of a problem for estimating

5      the total $CaCO_3$ MAR within the province, given that the rapidly-accumulating regions are more important for the overall burial flux, and therefore it is most important that they be well-constrained.

The heterogeneity uncertainty is independent of the others, and is addressed by the bootstrap resampling across multiple configurations of the province maps (Section B).





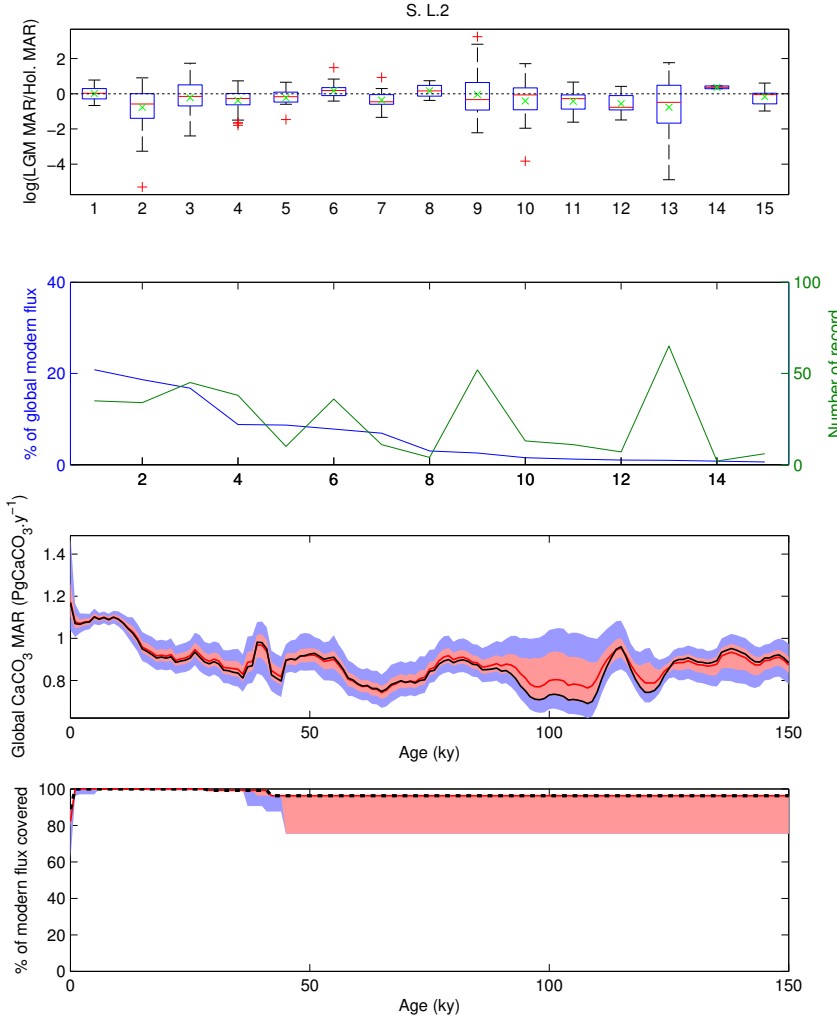

**Figure A2: Illustrative summary figure for the province-map S.L.2.** Top panel shows relative changes in downcore CaCO₃ MAR (ln(LGM/Holocene)) in each province. Box plot shows 0 % (bottom whiskers), 25 %, 50 % (median; red lines), 75 %, and 100 % (top whiskers) quantiles, as well as mean (green cross) and outliers (red cross), for each province. The second panel shows the contribution by province to modern CaCO₃ MAR, in descending order (blue curve) and the number of records used in each province (green). Panel 3 shows the history of absolute changes in CaCO₃ MAR for the last 150kyr, based on the bootstrap experiment (5[th], 25[th], 75[th] and 95[th] percentile ranges are





shaded, and the bootstrap mean plotted in red) compared to the calculation using all available records (black). Panel 4 shows the temporal variations in coverage, based on the bootstrap experiment. 5[th], 25[th],75[th] and 95[th] percentiles are shaded, and the bootstrap mean is plotted in red. Black dotted line corresponds to the coverage variations using all available records.

**A.5 Uncertainty in map reconstructions**

Inaccuracy in the spatial distribution of $CaCO_3$ burial in the modern map reconstruction will incorrectly weight the relative changes in different provinces in calculating the global burial flux. This uncertainty is independent of the others.
We discussed this aspect above (Fig. A2 and Table 1) and include it in the bootstrap below by repeating the calculations with the two available maps. Given that the two maps are constrained with the same data, this is likely an underestimate of the
10 true uncertainty. In the future, this could be revisited when new data-constrained maps of $CaCO_3$ burial flux become available. Nonetheless, our results indicate very similar temporal patterns for the two modern maps used, suggesting that our reconstruction is relatively insensitive to this uncertainty (Table 1). Given that this uncertainty is independent of the others, it is not likely to be highly significant for the final outcome.

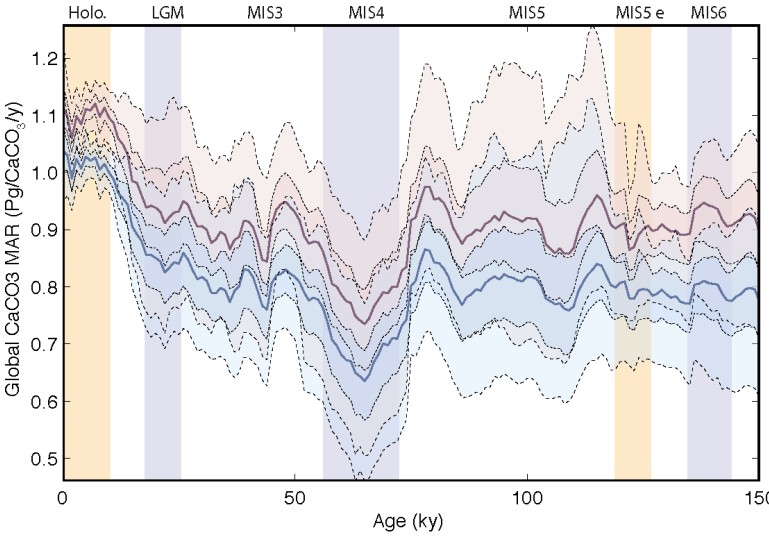

**Figure A3: Global carbonate burial using two different modern MAR maps. Lines and shaded areas show the reconstructed time histories of global carbonate burial using a modern carbonate MAR map based on sedimentary data (red) and on a metamodel tuned to fit the sedimentary data (blue). See also Fig. 2. The distribution shown by**





**dotted lines and shaded areas (5 th, 25 th, 50 th, 75th and 95th percentiles), was calculated from a bootstrap experiment performed on the 20 province maps used in the study.**

### B. Quantifying the uncertainties

In order to quantitatively estimate the impact of uncertainties itemized above on our reconstruction, we performed a series of

bootstrap experiments and the 20 province map set. Each experiment consisted of repeating the entire calculation for each province-map by randomly selecting $n$ records out of the $n$ available records in each province, allowing a single record to be selected multiple times. Each record was expressed as the relative deviation over time from the mean Holocene value. The geometric mean of the new set of $n$ time series was subsequently multiplied by modern MAR in the province, at each time step. For any time period in which no downcore records were available within a province, the MAR for that province was set

at the modern value. The coverage was calculated at each time step, as the sum of the modern MAR for provinces where downcore records were available, divided by the global modern MAR. This operation was repeated 10000 times for each province-map.

We also tested the influence of the modern MAR map by using the two modern MAR maps shown in Fig. 2. These two maps have different absolute modern $CaCO_3$ MAR and also distribute the modern MAR differently among provinces

(Table1), allowing a test of the robustness of our method to different modern MAR distributions (Fig. A3). The relative changes in our reconstructed global carbonate burial over time do not depend on the choice of maps, as shown in Fig. A3.

The results of the bootstrap experiment for each province-map, using modern map 1, are shown in Fig. A3, and a summary of the results is displayed in Fig. A3. It is important to keep in mind that the coverage calculated here can only be equal or lower than the normal calculation using all records, due to bootstrap resampling (Fig. A2, panel 4).

The province-map scenarios that result in the lowest amplitude during MIS2 and MIS4 are also the ones that have the lowest coverage, as a higher proportion of the total burial is set as constant for these maps (see Table 3, Fig. A5). This effect also explains the lower amplitude affecting MIS6-MIS5 transition as compared to MIS2-MIS1. Using the mean and the standard deviation of the 20 province maps scenarios, which do not cover exactly the same surfaces, tends to decrease the variability of the mean curve while increasing the standard deviation such as the final distribution of the burial is rather conservative.




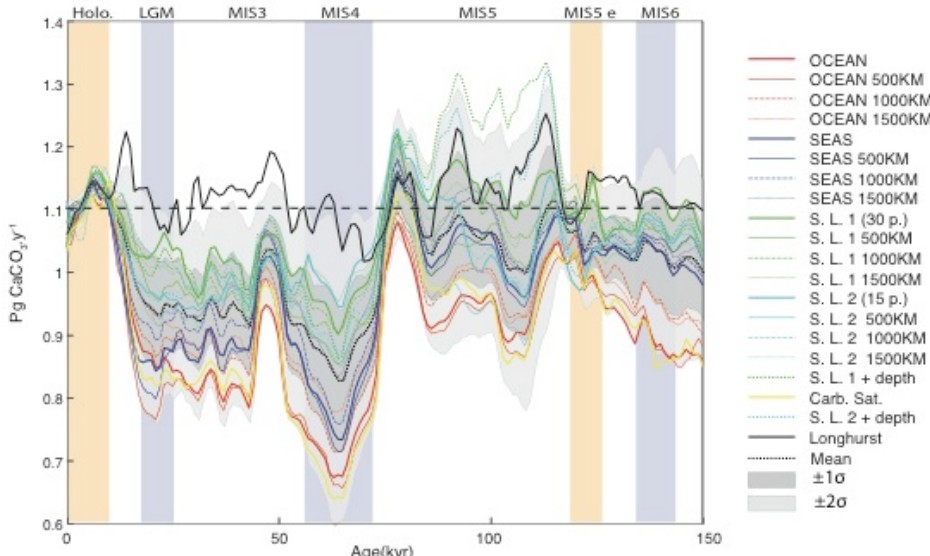

**Figure A4: Detail of Fig. 6, panel f, corresponding to the final reconstruction.**




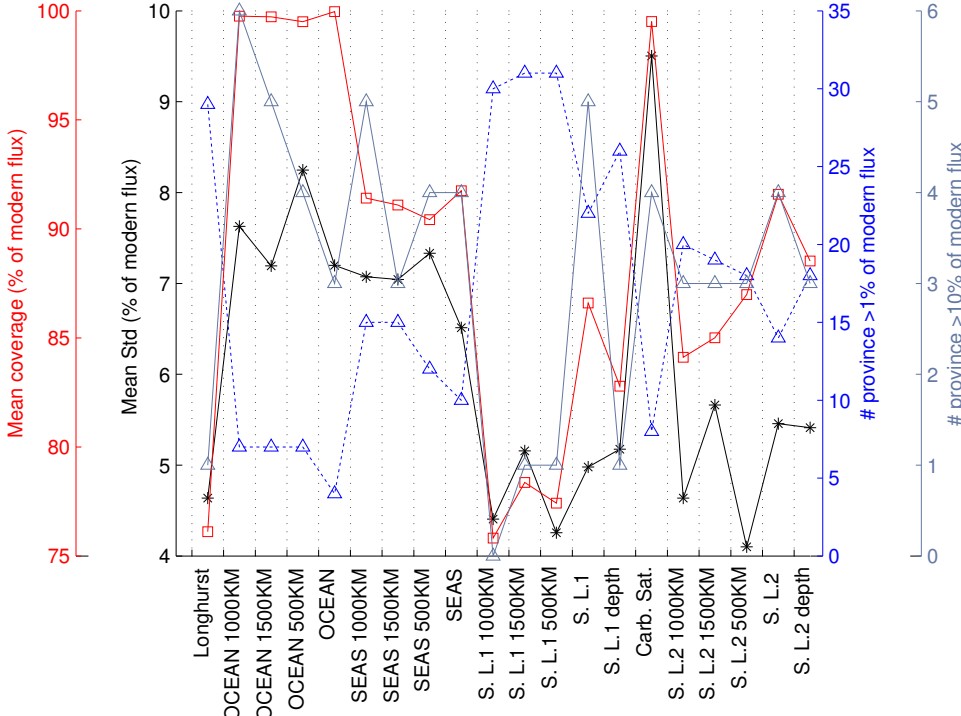

**Figure A5: Summary of province-map experiments. Black symbols show the mean of the standard deviation at each time step of the bootstrap experiment, for each province-map. Red symbols show the mean coverage for the bootstrap experiment, for each province-map. The number of provinces accounting for more than 1% and 10% of the modern flux for each province-map are shown by the blue and grey symbols, respectively.**

Although we could use these bootstrap reconstructions to provide a mean and standard deviation of the forcings, we instead use the timeseries using all available records, in order to make as complete a use as possible of the available data. Given that the individual province-map scenarios are distributed quite symmetrically around the mean scenario (Fig. A4 and A6), we used the relative changes in the mean reconstruction as shown in Fig. A6 panel A, to which we add, at each time steps, a normally distributed fraction of the relative standard deviations calculated from the 20 province-map scenarios. Note that the mean standard deviation used here (equal to the mean of the mean standard deviation calculated from the 20 province maps scenarios at each time step, equivalent to approximately 7 % of the modern global burial flux) is in the higher range of the mean standard deviation calculated from the bootstrap experiment for each province map (Fig. A5). Thus, although we do

not use the bootstrap results here directly, they are consistent with the uncertainty range applied. The timeseries are prescribed as variations relative to the modern global burial flux (Fig A6, panel B and C).

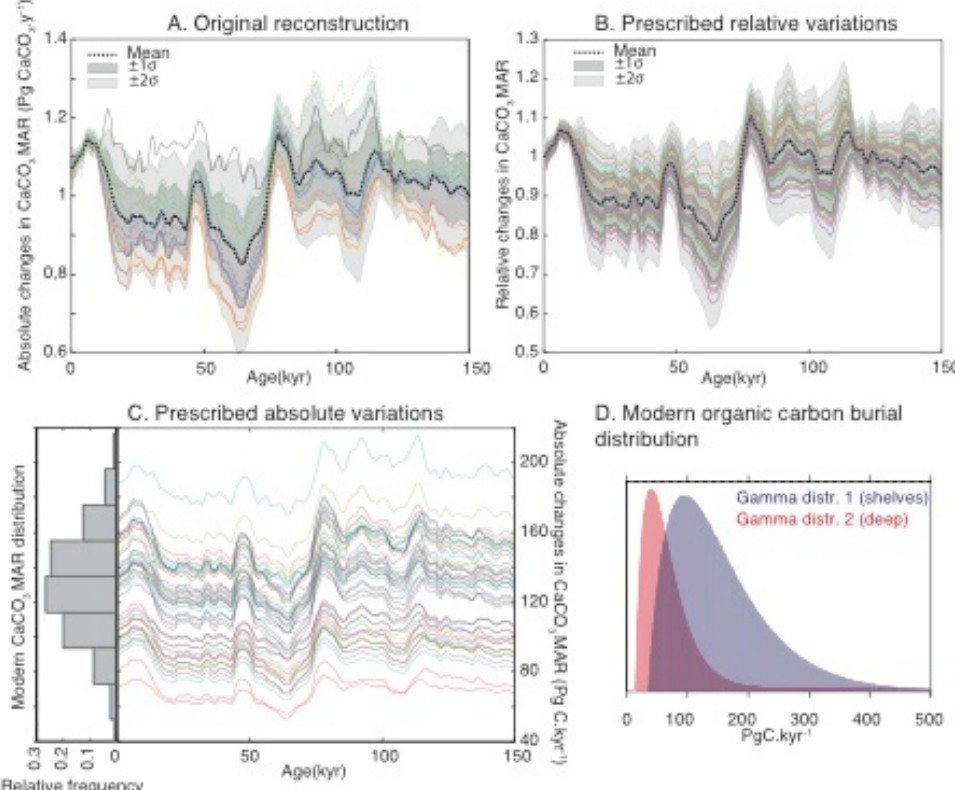

**Figure A6: Construction of time series for scenarios. A. Original reconstruction from the 20 province maps scenarios. B. Prescribed relative variations derived from the distribution in A. C. Prescribed absolute variations derived from the product of B and the assigned modern burial distribution (see details in Table 3). D. Prescribed modern organic carbon burial distribution over shelf (gamma distr. 1, mean=160; median=140; mode=100; std=80) and permanently submerged deep-sea environments (gamma distr. 2, mean =70; median =60; mode =40; std =40).**



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
