# Peer review of "Carbon burial in deep-sea sediment and implications for oceanic inventories of carbon and alkalinity over the last glacial cycle"

_Climate of the Past, 2018_

## Referee Comment (RC1) · A. Schmittner (Referee) · 14 Jun 2018

The authors reconstruct changes in global CaCO3 burial fluxes through the last glacial cycle using a large number of ocean sediment core data and evaluate its effects on global DIC and alkalinity changes and on d13C. They also evaluate effects of shelf burial of CaCO3 and organic carbon by reducing the fluxes proportional to the shelves surface area. They conclude both changes in deep ocean and shelf burial did potentially affect global DIC, ALK and d13C.

[Figure]

I think this is a very nice paper with important implications for the understanding of glacial-interglacial changes in the carbon cycle. It is very well written (except for a few typos) and nicely illustrated and it is a great contribution to CP.

In my opinion the most important contribution of this paper is a quantitative reconstruction of deep ocean CaCO3 burial fluxes. However, the effects of these on deep ocean DIC, ALK and d13C are relatively small (Fig. 8A). On the other hand, shelf burial would have larger effects on DIC, ALK and d13C (Fig. 8B) but it is not reconstructed based on data but rather based on the assumption of fluxes proportional to shelf surface area. I wonder how good this assumption is. E.g. I could imagine that most burial happens on the inner shelf and is not distributed equally across the shelf area (defined as depth < 100 m). If this is the case, then the burial fluxes may not have decreased that dramatically. I don't know if what I think is correct, but it seems to me that the shelf burial changes are more uncertain and less constrained by observations than the deep burial changes. Perhaps the authors want to entertain this thought in their discussion.

Also, I think recent papers by Wallmann and collaborators already addressed the effects of sea level changes in shelf burial and d13C. This should be acknowledged.

P1, L15: "... removal rates, which mainly occurs in marine sediments" I think the correct syntax should be "occur". However, I'm not sure this statement is correct. If the "active" carbon inventory include terrestrial soils and vegetation and the "inactive" or "geological" inventory includes permafrost and peats, then the fluxes between the active and inactive land reservoirs may also be important at least during certain periods.

P1, L17-18: I don't think this sentence is supported by the evidence presented. I think what you wanted to say was something like " ... the reconstruction provides a first order constraint on the effect of changes in deep-sea burial fluxes on carbon and alkalinity inventories over the last glacial cycle." I think this ("the effect of changes in deep-sea burial fluxes on") qualifier is needed because you don't provide constraints on the absolute (total) DIC and ALK changes, just those resulting from changes in

burial fluxes.

P1, L21: "active carbon inventory" It is not clear what this is. Does it include ocean sediments? I wonder what you wanted to say with this sentence. If the active inventory includes atmosphere, ocean and land then I think it is not news that it was a dynamic, interactive component of glacial cycles.

Fig. 1: Please explain why three arrows are red.

P4, L20: Parenthesis should be after "Milliman"

Fig. 2: Consider using a color scheme readable to color-blind people (without red or green)

P6, L8: The figure says 100-150.

P6, L13-15: I thought the d13C of buried CaCO3 was close to that of surface DIC assuming that most of the CaCO3 was produced at the surface.

P6, L30: Parenthesis should be after "Burdige"

P7, L7: Parenthesis

P9, L12: It is claimed here that the province approach is preferable. Has this actually been shown somewhere? Why could it not also be prone to interpolation and extrapolation biases?

P9, L17: Remove parenthesis with Cartapanis. Typo: it should be "assumes" instead of "assume"

P10, L13: the noaa ftp link was not working

P10, L20, 21: Please report time periods used for Holocene and LGM.

P11, L2: What are the provinces?

P11, L22: The assumption of constant shallow/deep partitioning is most likely not valid.

E.g. the d13C data suggest more DIC in the deep ocean and thus a larger surface to deep DIC gradient during the LGM. What are the consequences of this assumption for the results?

Fig. 3: What are the different lines? Labels OCEAN, SEAS, S. L. 1, S. L. 2?

P12, L1-2: The fluxes between land and the ocean-atmosphere are neglected. What could be the consequences of this assumption?

P12: Please provide the equations for the mass balance calculations.

Table 2: Why do the numbers not correspond to those in Fig. 1?

Fig. A1: Please include the numbers in the panels. I assume that 1 is the upper left and 4 is the upper right.

P13, L8: In the title please specify with MAR you're considering. I think it is CaCO3 MAR in the deep ocean, right? As opposed to CaCO3 MAR in the shallow ocean.

P15, L 7: Please provide definition of MIS5e.

Fig. 5: Label in panel E has a typo "buk" should be "bulk"

P18, L9: Syntax: replace "in" (first word) with "of"

P20, L10: Typo: "in put" should be "input"

P26, L5: I think the reference to Fig. 1 may be wrong.

P26, L15: Is there evidence for an enhanced soft tissue pump at the start of MIS4?

P30, L9: I think the first who has suggested to use deep ocean d13C to reconstruct terrestrial carbon biomass was Shackleton (1977), Carbon-13 in Uvigerina: Tropical rainforest history and the Equatorial Pacific carbonate dissolution cycles, in The Fate of Fossil Fuel CO2 in the Oceans, edited by N. R. Andersen and A. Malahoff, pp. 401–427, Plenum, New York.

[Figure]

P30, L12-14: This is an important conclusion, but it has already been suggested by Wallmann et al., (2016, Clim. Past., page 349). BTW this reference is listed twice in the references section.

P30, L27: Where is this shown? Please refer to figure.

P31, L7: Who and how was the quality control done?

P31, L13-14: "due to enhanced soft tissue pump" I think it would be better to remove this attribution since other processes such as disequilibrium, solubility may also have been responsible for the reduction in carbonate burial.

———————————————

---

## Referee Comment (RC2) · Anonymous Referee #2 · 19 Jun 2018

The manuscript of Cartapanis et al. is a very interesting study that contains a tremendous effort on compiling and standardising paleoceanographic data from across the global ocean. In this study the authors use this dataset to eventually estimate the global burial flux of carbonate over the last glacial cycle. Their main results indicate that carbonate burial in the deep-sea across the last interglacial was similar to the Holocene but that it decreased during the glacial until MIS 3, and how this fact would have had an impact in the DIC, ALK and d13C inventories across the glacial. In order to achieve their final conclusions, the authors divide the ocean into 20 provinces and use their MAR calculations to simulate 8 possible scenarios isolating the impact of several variables. This study highlights that carbonate burial fluxes played a role

in glacial-interglacial carbon cycle changes and therefore the potential importance of taking them into account in any further calculation/interpretation.

This is a very relevant and well-performed study very useful for the whole paleoclimatologic community. However, I have found relatively hard to go through the whole document, fundamentally due to its organization and length, which in my view affects the clarity of the main messages. Obviously, I completely understand that to capture such a big job and complicated topic in a single document is not an easy task to accomplish, but I find that the manuscript in general, and several sections in particular, might perhaps be shortened. The feeling sometimes is that this study might be split in two documents. The Results section, for instance, might well be the Discussion section in a more methodological paper, since there is indeed a lot of discussion about the results rather than a simple description of them. I would suggest that the authors consider transferring some of the results to the discussion section, and not only leave it for the more strict paleoceanographic interpretations. My recommendations are therefore mostly related to the format.

- In the Introduction, section 1 leaves clear the current situation and the purpose of the study, and the review of sources and sinks in section 2 supplies with very useful data. However, some of the sentences within and between paragraphs in this section 2 seem to expose a large amount of information but without a clear purpose and not being entirely connected. Some of my suggestions, a part from trying to shorten these subsections in general, would be to separate paragraphs in clear groups of ideas and link the sentences with more connectors, as well as perhaps to state clearly what the general current 'agreements' in each subsection are and then go into the details of each study.

- For Methods and Results, I would suggest to make a complete scheme of the methodology applied (including the analysis performed in the results section) followed by the correspondent explanations. I would also highly recommend shortening both sections and I think it would be very useful to start the Results with a brief summary of what is

going to be told, as well as with a complete explanation of all the experiments that have been performed and how they link with the plots in Figures 7, 8 and 10 (it is mentioned in the caption of Table 2 in methods, but I think it would help to explain all this straight and clear in results).

- Regarding the Discussion, as I said before, I think it would be useful to transfer part of the results into the discussion section.

- When referring to the Appendices, the authors should try to consistently be specific.

- A more clear explanation about the connection between the experiments and the final results might be stated.

- It seems confusing why the authors talk about organic carbon burial in the final conclusions of the study as main results. Are not those results from Cartapanis et al. 2016?

- The authors should explain more clearly how particular changes in the Atlantic vs pacific in figure 11 (or even in figure 6) could be inferred.

- There are, sometimes, quite long and complicated sentences that might be rephrased for better understanding, e.g. line 23 page 10, line 31 page 12, line 1 page 26, line 13 page 31, etc.. In the same way, I personally find the use of commas excessive making the reading harder.

- I would consistently use Corg and CaCO3, as well as DIC and ALK across the manuscript, as it might allow the reader to rapidly identify what the authors are referring to.

- I would avoid starting sections with 'Moreover' or 'However', like in sections 5.2 and 5.4, respectively. In my view, those connectors are used to either add or rebate information that has been mentioned in the same paragraph or in the previous one, but not in another section. I think those sections should either be together with the previous ones, or include an initial brief summary highlighting the reason why they are going to

be discussed once again.

Some technical mistakes spotted:

- Section 3.5 should be numbered as 3.4.

- Line 23 page 31: proper references are missing.

- $CO_3^-$ is sometimes typed as $CO_3^2$ or $CO_3^2=$. It should be corrected.

- The titles of sections 5.2 and 5.4 have typos, as well as lines 16 and 20 in page 31.

- There are at least a couple of negative contractions that should be corrected; line 14 page 31 and line 23 page 26.

---

## Author Comment (AC1) · 24 Jul 2018

Response to interactive comments on "Carbon burial in deep-sea sediment and im­plications for oceanic inventories of carbon and alkalinity over the last glacial cycle" (cp-2018-49) by A. Schmittner (Referee ; https://doi.org/10.5194/cp-2018-49-RC1).

The authors reconstruct changes in global CaCO3 burial fluxes through the last glacial cycle using a large number of ocean sediment core data and evaluate its effects on global DIC and alkalinity changes and on d13C. They also evaluate effects of shelf burial of CaCO3 and organic carbon by reducing the fluxes proportional to the shelves surface area. They conclude both changes in deep ocean and shelf burial did poten-

tially affect global DIC, ALK and d13C. I think this is a very nice paper with important implications for the understanding of glacial-interglacial changes in the carbon cycle. It is very well written (except for a few typos) and nicely illustrated and it is a great contribution to CP. In my opinion the most important contribution of this paper is a quantitative reconstruction of deep ocean CaCO3 burial fluxes. However, the effects of these on deep ocean DIC, ALK and d13C are relatively small (Fig. 8A). On the other hand, shelf burial would have larger effects on DIC, ALK and d13C (Fig. 8B) but it is not reconstructed based on data but rather based on the assumption of fluxes proportional to shelf surface area. I wonder how good this assumption is. E.g. I could imagine that most burial happens on the inner shelf and is not distributed equally across the shelf area (defined as depth < 100 m). If this is the case, then the burial fluxes may not have decreased that dramatically. I don't know if what I think is correct, but it seems to me that the shelf burial changes are more uncertain and less constrained by observations than the deep burial changes. Perhaps the authors want to entertain this thought in their discussion. Also, I think recent papers by Wallmann and collaborators already addressed the effects of sea level changes in shelf burial and d13C. This should be acknowledged.

-We thank the reviewer for his supportive comments on the quality and importance of our manuscript. It is absolutely true that the burial on the shelves remains relatively poorly constrained in the present and in the past, despite its potential importance in balancing the alkalinity budget of the ocean. For that reason, we used a very simple but illustrative scenario based on the commonly-assumed reduction of shelves burial during glacial times, but take into account the uncertainties in modern burial based on published literature.

-We recognize that the uncertainties affecting this component of the global flux did not come across strongly enough in the original manuscript, and will emphasize them in the revised manuscript, pointing out the necessity to address this knowledge gap. The Wallmann papers will be cited more prominently in the appropriate section of the

manuscript.

-We are also thankful for the corrections of few typos, which have been corrected in the final manuscript.

P1, L15: "... removal rates, which mainly occurs in marine sediments" I think the correct syntax should be "occur". However, I'm not sure this statement is correct. If the "active" carbon inventory include terrestrial soils and vegetation and the "inactive" or "geological" inventory includes permafrost and peats, then the fluxes between the active and inactive land reservoirs may also be important at least during certain periods.

-We agree with this statement and have redefined the 'active pool' to include permafrost and peat (but not marine sediments).

P1, L17-18: I don't think this sentence is supported by the evidence presented. I think what you wanted to say was something like " : : : the reconstruction provides a first order constraint on the effect of changes in deep-sea burial fluxes on carbon and alkalinity inventories over the last glacial cycle." I think this ("the effect of changes in deep-sea burial fluxes on") qualifier is needed because you don't provide constraints on the absolute (total) DIC and ALK changes, just those resulting from changes in burial fluxes.

-We agree with this statement and have reformulated this sentence accordingly.

P1, L21: "active carbon inventory" It is not clear what this is. Does it include ocean sediments? I wonder what you wanted to say with this sentence. If the active inventory includes atmosphere, ocean and land then I think it is not news that it was a dynamic, interactive component of glacial cycles.

-We reformulated this sentence

Fig. 1: Please explain why three arrows are red.

-Red arrows represent alkalinity fluxes. The figure legend has been amended accordingly.

P4, L20: Parenthesis should be after "Milliman"

-Corrected.

Fig. 2: Consider using a color scheme readable to color-blind people (without red or green)

-We will consider it for future publication. We took care to use a perceptually uniform color map, but unfortunately did not sufficiently consider color-blind readers.

P6, L8: The figure says 100-150.

-Corrected.

P6, L13-15: I thought the d13C of buried CaCO3 was close to that of surface DIC assuming that most of the CaCO3 was produced at the surface.

-Yes, we agree. But this is not inconsistent with the quoted sentence.

P6, L30: Parenthesis should be after "Burdige" Corrected.

P7, L7: Parenthesis

-Corrected.

P9, L12: It is claimed here that the province approach is preferable. Has this actually been shown somewhere? Why could it not also be prone to interpolation and extrapolation biases?

-We agree that the phrasing regarding 'interpolation and extrapolation biases' was misleading, as it is true that our approach is also prone to these biases to some degree. However, to us it would appear obvious that the province approach would be preferable, given that it is more likely to take advantage of local covariation, and that we include many different province configurations in order to capture a range of different possible interpolation and extrapolation biases. For example, our 'Oceans' provinces

are at such large scale that they are relatively close to a standard interpolation.

-We would therefore propose to change the text in order to say that we believe the province approach is an improvement over straightforward interpolation, and that because it assumes local covariance among oceanographically/geographically similar regions, it is less likely to produce spurious results. Future work could explore the validity of the approach more quantitatively.

P9, L17: Remove parenthesis with Cartapanis. Typo: it should be "assumes" instead of "assume"

-Corrected.

P10, L13: the noaa ftp link was not working

-Thanks for pointing this out – It was impossible to include the link into a PDF document for some reason.

P10, L20, 21: Please report time periods used for Holocene and LGM.

-Done.

P11, L2: What are the provinces?

-We reformulated the sentence

P11, L22: The assumption of constant shallow/deep partitioning is most likely not valid. E.g. the d13C data suggest more DIC in the deep ocean and thus a larger surface to deep DIC gradient during the LGM. What are the consequences of this assumption for the results?

-This assumption could slightly impact the d13C of the carbon bearing compounds formed at the surface, thus the isotopic composition of the flux out of the system. However, the difference is on the order of a few tenths of a permil, which is much smaller than the uncertainty in the organic carbon d13C composition, which may also

change with pCO2 and growth rates. This small effect would not have a significant impact on the results.

Fig. 3: What are the different lines? Labels OCEAN, SEAS, S. L. 1, S. L. 2?

-These lines correspond to the different provinces scenarios as described in figure A1. They will be clarified in the revised caption.

P12, L1-2: The fluxes between land and the ocean-atmosphere are neglected. What could be the consequences of this assumption?

-This is a good point. Our assumption is roughly equivalent to constant fluxes, though we will modify the text in order to add a caveat about this.

P12: Please provide the equations for the mass balance calculations.

-The equation has been provided in the manuscript

Table 2: Why do the numbers not correspond to those in Fig. 1?

-Thanks for pointing this out, the deep TOC was not updated from a prior estimate in the figure: corrected!

Fig. A1: Please include the numbers in the panels. I assume that 1 is the upper left and 4 is the upper right.

-Amended.

P13, L8: In the title please specify with MAR you're considering. I think it is CaCO3 MAR in the deep ocean, right? As opposed to CaCO3 MAR in the shallow ocean.

-Corrected.

P15, L 7: Please provide definition of MIS5e.

-We have defined MIS5e as the period 119 – 124 ka, according to (Lisiecki and Raymo, 2005).

Fig. 5: Label in panel E has a typo "buk" should be "bulk"

-Fixed.

P18, L9: Syntax: replace "in" (first word) with "of"

-Corrected.

P20, L10: Typo: "in put" should be "input"

-Corrected.

P26, L5: I think the reference to Fig. 1 may be wrong.

-Corrected

P26, L15: Is there evidence for an enhanced soft tissue pump at the start of MIS4?

-Many studies have given evidence for a reduce deep circulation at mis4 which induces stronger soft tissue pump in the sense of more DIC stored at depth, concomitant with clue for enhanced production and export (see details in (Kohfeld and Chase, 2017))

P30, L9: I think the first who has suggested to use deep ocean d13C to reconstruct terrestrial carbon biomass was Shackleton (1977), Carbon-13 in Uvigerina: Tropical rainforest history and the Equatorial Pacific carbonate dissolution cycles, in The Fate of Fossil Fuel CO2 in the Oceans, edited by N. R. Andersen and A. Malahoff, pp. 401– 427, Plenum, New York.

-Thanks for this, reference added.

P30, L12-14: This is an important conclusion, but it has already been suggested by Wallmann et al., (2016, Clim. Past., page 349). BTW this reference is listed twice in the references section.

-We have added the citation to Wallmann for this conclusion.

P30, L27: Where is this shown? Please refer to figure.

-Amended

P31, L7: Who and how was the quality control done?

-We have added a reference to the description of the quality control in section 3.3, rather than in the conclusion, which refers to (Cartapanis et al., 2016)

P31, L13-14: "due to enhanced soft tissue pump" I think it would be better to remove this attribution since other processes such as disequilibrium, solubility may also have been responsible for the reduction in carbonate burial.

-Good point, we have removed this aspect.

––––––––––––––––––––––––––––

---

## Author Comment (AC2) · 24 Jul 2018

Response to interactive comments on "Carbon burial in deep-sea sediment and implications for oceanic inventories of carbon and alkalinity over the last glacial cycle" (cp-2018-49) by referee 2 (https://doi.org/10.5194/cp-2018-49-RC2).

The manuscript of Cartapanis et al. is a very interesting study that contains a tremendous effort on compiling and standardising paleoceanographic data from across the global ocean. In this study the authors use this dataset to eventually estimate the global burial flux of carbonate over the last glacial cycle. Their main results indicate that carbonate burial in the deep-sea across the last interglacial was similar to the

[Figure]

Holocene but that it decreased during the glacial until MIS 3, and how this fact would have had an impact in the DIC, ALK and d13C inventories across the glacial. In order to achieve their final conclusions, the authors divide the ocean into 20 provinces and use their MAR calculations to simulate 8 possible scenarios isolating the impact of several variables. This study highlights that carbonate burial fluxes played a role in glacial-interglacial carbon cycle changes and therefore the potential importance of taking them into account in any further calculation/interpretation. This is a very relevant and well-performed study very useful for the whole paleoclimatologic community. However, I have found relatively hard to go through the whole document, fundamentally due to its organization and length, which in my view affects the clarity of the main messages. Obviously, I completely understand that to capture such a big job and complicated topic in a single document is not an easy task to accomplish, but I find that the manuscript in general, and several sections in particular, might perhaps be shortened. The feeling sometimes is that this study might be split in two documents. The Results section, for instance, might well be the Discussion section in a more methodological paper, since there is indeed a lot of discussion about the results rather than a simple description of them. I would suggest that the authors consider transferring some of the results to the discussion section, and not only leave it for the more strict paleoceanographic interpretations. My recommendations are therefore mostly related to the format.

-We thank the reviewer for his supportive comments on the quality of our study and the effort required to achieve such analyses. We also recognize that the huge amount of information required to properly describe our work can sometimes obfuscate our message, and that it has been quite a challenge to keep the length of our study sufficiently short to fit into a single paper. But we think that the combination of data based estimates and model derived scenarios is the best option to fully show the relevance of our study.

In the Introduction, section 1 leaves clear the current situation and the purpose of

the study, and the review of sources and sinks in section 2 supplies with very useful data. However, some of the sentences within and between paragraphs in this section 2 seem to expose a large amount of information but without a clear purpose and not being entirely connected. Some of my suggestions, a part from trying to shorten these subsections in general, would be to separate paragraphs in clear groups of ideas and link the sentences with more connectors, as well as perhaps to state clearly what the general current 'agreements' in each subsection are and then go into the details of each study.

-In order to facilitate reading of our manuscript, we added subtitles in section 2 and slightly modified the paragraphs and text as suggested.

For Methods and Results, I would suggest to make a complete scheme of the methodology applied (including the analysis performed in the results section) followed by the correspondent explanations. I would also highly recommend shortening both sections and I think it would be very useful to start the Results with a brief summary of what is going to be told, as well as with a complete explanation of all the experiments that have been performed and how they link with the plots in Figures 7, 8 and 10 (it is mentioned in the caption of Table 2 in methods, but I think it would help to explain all this straight and clear in results).

-We recognize that the original organization of the manuscript wasn't optimal. We have now reorganized the material and method and results sections. The CaCO3 reconstruction section and modeling aspects are now more clearly organized and separated.

Regarding the Discussion, as I said before, I think it would be useful to transfer part of the results into the discussion section.

-We agree, and hope that the reorganization of the manuscript will help with readability.

When referring to the Appendices, the authors should try to consistently be specific.

-We corrected the references to the appendices.

A more clear explanation about the connection between the experiments and the final results might be stated. It seems confusing why the authors talk about organic carbon burial in the final conclusions of the study as main results. Are not those results from Cartapanis et al. 2016?

-Here, we used the deep-sea organic carbon burial reconstruction from Cartapanis et al. 2016 but we also include the Corg burial on shelves, which was not included in Cartapanis et al. 2016. The potential impact of Corg burial on shelves has significant implications for the global mean oceanic d13C value, given that the estimated shelf fluxes are larger than those of the deep sea.

The authors should explain more clearly how particular changes in the Atlantic vs pacific in figure 11 (or even in figure 6) could be inferred.

-We have more clearly addressed this particular aspect in the revised manuscript (p. 33 L7)

There are, sometimes, quite long and complicated sentences that might be rephrased for better understanding, e.g. line 23 page 10, line 31 page 12, line 1 page 26, line 13 page 31, etc.. In the same way, I personally find the use of commas excessive making the reading harder.

-We corrected some of the sentences that appeared clumsy and tried to avoid the use of commas where adequate.

I would consistently use Corg and CaCO3, as well as DIC and ALK across the manuscript, as it might allow the reader to rapidly identify what the authors are referring to.

-Corrected

I would avoid starting sections with 'Moreover' or 'However', like in sections 5.2 and 5.4, respectively. In my view, those connectors are used to either add or rebate information that has been mentioned in the same paragraph or in the previous one, but not in

another section. I think those sections should either be together with the previous ones, or include an initial brief summary highlighting the reason why they are going to be discussed once again.

-Corrected

Some technical mistakes spotted: Section 3.5 should be numbered as 3.4. Line 23 page 31: proper references are missing. $CO_3^{2-}$ is sometimes typed as $CO_3^{2}$ or $CO_3^{2=}$. It should be corrected. The titles of sections 5.2 and 5.4 have typos, as well as lines 16 and 20 in page 31. There are at least a couple of negative contractions that should be corrected; line 14 page 31 and line 23 page 26.

-Corrected

---

## Author Comment (AC3) · 24 Jul 2018

[revised manuscript text omitted]

**4.4. Combined impact of sedimentary burial and carbon input variations**

It has been suggested that the rate of carbon input to the active pool from volcanic sources increased during glacial maxima (Lund et al., 2016; Ronge et al., 2016) and/or deglaciations (Huybers and Langmuir, 2009; Stott et al., 2009). For the purpose of this discussion, we consider peatlands and permafrost part of the 'active' pool, but not surficial marine sediments. The magnitude of these carbon input variations is highly uncertain, and reconstructions of relative changes in volcanic outgassing rates over time suggest that the volcanic carbon input was notably delayed, relative to the deglacial increase in pCO$_2$ recorded in ice cores, requiring other mechanisms to be important at this time (Huybers and Langmuir, 2009; Roth and Joos, 2012). Nonetheless, it would appear likely that changes in geological carbon inputs over the glacial cycle were significant. We do not attempt to provide an exhaustive test of the existing hypotheses for temporally-variable carbon input, nor do we consider peat and permafrost changes which could have played important additional roles (Zech, 2012). 
[revised manuscript text omitted]

---

## Editor Comment (EC1) · L. Skinner (Editor) · 14 Sep 2018

Dear Olivier,

I must first apologise for the time it has taken to process this manuscript. To some extent, this has stemmed from the stamina levels required to read through the weighty manuscript from 'cover to cover'; it contains a lot! I concur with the reviewers that this is a very interesting and indeed important contribution, though I also concur that this relevance and importance is somewhat obfuscated by the text and its organisation. Of course, your revisions have helped significantly to address this last issue; however, having gone through the manuscript myself a number of times now, I find that there are

still outstanding 'editorial' issues that could be improved. I do not wish to delay much further the publication of this study; however, I feel that it will find a wider readership and will make a greater impact if it can be polished a little more. In order to aid in this (i hope), I provide below a list of issues that I think are worth considering. I would encourage you to address all of these comments in a revised manuscript, to be accompanied by a brief commentary (the use of tracked changes as previously will allow you to keep the commentary short).

Once again, thank you for submitting this very interesting study to Climate of the Past, and for your patience in arriving at a final version that I hope will do full justice to the content of your study.

Sincerely, Luke Skinner

Outstanding matters to address or consider:

P.4, line 3: some of these references refer to peat/permafrost carbon, yet you define these reservoirs as being part of the 'active' carbon pool, rather than the 'geological' carbon pool. Please reconcile, or clarify.

P.4, line 13: Perhaps: "...allows for some first order estimates of changes in... These estimates suggest that large changes in both carbon and...."

P.4, line 18: I think at this point, given the length and complexity of the manuscript, it might be useful to provide a sort of 'key' to what follows, e.g. " In section 2 we provide an overview of.... In section 3 we present our reconstruction... and methods... In section 4 we describe some modelling scenarios... etc..."

P.4, line 24: "These quantitative constraints... past changes in the global carbon and alkalinity budgets."

P.5, line23: shells and shell fragments? Should we describe coccoliths as shells?

P.5, line 26: "...with increasing pressure and therefore water depth."

P.9, line 3: In general, when referring to carbon, please be specific as to organic carbon, carbonate, or both.

P.10, line 10: Are you saying that the most recent rapid sea level drop was at the start of the last deglaciation? Please correct or clarify.

P.10, line 13: there has been a recent proposal that this value might be more positive, closer to arc volcanoes at -3 permil (Mason et al., 2017). Is this worth noting? Does it make a difference to your calculations?

P.11, line 1: pleas change to 'the glacial' or 'glacials'.

P.11, line 2: ditto. . .

P.11, line 4: Please break up the sentence. E.g. ". . .although this too has been debated (von Blanckenburg et al., 2015). It is also possible that. . . lower temperature, offsetting any increase in physical erosion and allowing global erosion to remain relatively stable. . ."

P.11, line 6: Note also some new perspectives on the glacial weathering and the temperature feedback (e.g. Torres et al., PNAS 2017).

P.12, line 10: I suggest to formulate this as a 'bulleted' list, allowing the intervening sentences: 1). . .; 2). . .; and 3)

P.12, line 21: assumes.

P.12, line 22: replace nor with or. I think you can cut "which would be expected to alter etc. . ." Perhaps instead at this stage it is worth making a brief note on the complicating effects of a world always out of equilibrium, contra the approach adopted here?

P.14, line 19: ". . .(no change), while. . .."

P.21, lines 10-21: this whole paragraph is very unclear. Please have a go at clarifying what is meant.

[Figure]

P.22, line 9: On P.18, line 6 it is stated that "we chose to correct for compaction.. by assuming.. global bulk burial rate was equal during the late Holocene and MIS5e…" It seems wrong to then 'report' that mean deep sea carbonate burial was not different form modern during MIS5e, or have I missed something? Please correct or clarify.

P.23, line 3: The phrasing here is not optimal I think; please consider re-phrasing.

P.23: Here is a more substantial issue: the 'model' is not adequately described. Indeed, one of the reviewers asked for equations, and this has not been addressed as far as I can tell. The equation that is provided simply aims at describing the general notion of mass balance, though it needs to be corrected since a rate of change of concentration [C] cannot be equal to the difference between two concentrations; different symbols/nomenclature must be used to distinguish [C] from d[C]/dt, and to include the relevance of reservoir size etc... Furthermore, it is important that the air-sea partitioning scheme and shallow/deep partitioning scheme are fully described, and that equations for the rate of change of e.g. [C] in the atmosphere, shallow and deep ocean etc… be provided.

P.23, line 16: please consider rephrasing, e.g.: "…we have not considered variable erosion fluxes, due to a paucity of consistent observational constraints on these…"

P.24: I think that at this stage in the manuscript it would be useful to provide a short overview of the various scenarios that are considered and why (i.e. of what is to come in the next subsections). E.g. "Next we consider X different model scenarios where we calculate XYZ, on the basis of assumed inputs XYZ etc etc…. We label/name each of these scenarios KJH, as listed in table 3… Our goal with these scenarios is to… "

P.25: Is it possible to clearly link each of these sub-sections to the relevant model scenarios, e.g. in the title, or in the first few lines? I think it would be helpful.

P.25, line 12: Please add another word or two to this line to specify why the minimum is a robust outcome of the CaCO3 burial fluxes. The minimum is close to the minimum

[Figure]

in carbonate/ALK output, which would be counter-intuitive given constant input fluxes, though I suspect that the minimum in ALK in fact occurs just before the minimum in carbonate burial (MIS4) and results from the sustained high/peak outputs leading up to the MIS5/4 transition. Please clarify.

Fig. 7: In this figure (and the others) please identify the vertical shaded bars. Also, it might be useful to clarify that 'net C source' is the net flux of carbon from the geological to the active carbon reservoirs, as defined here... if indeed this is correct.

Fig. 7, caption: I think it would be clearer to write: "A. Variable deep sea CaCO3.. B. Variable deep dea.." etc.. The same goes for the other figures.

P.26, line 3: "...experiments, the mean.."

P.26, line 4: is it worth noting by how much 500PgC exceeds the modern flux, and that this would imply a downward drift in atmospheric CO2 across a glacial cycle if it was not actually realised?

P.32, line 8: This is a sentence fragment; please amend. E.g. "Although the preserved... (appendix A), our broad spatial...."

P.32, line 12: remove 'then'... Though I think this could be stated much more clearly with a different construction: e.g. "Absolute changes in CaCO3 burial were obtained by scaling/multiplying reconstructed relative changes in CaCO3 MAR to/by the modern MAR. Uncertainties were estimated..."

P32, line 17: is 'very large' necessary?

P.32, line 18: again, was it not assumed a priori that MIS5 CaCO3 burial was the same as modern/late Holocene? Hence, for example: "on the assumption that it was the same during MIS5 and the late Holocene, CaCO3 burial dropped to the lowest rates...".

P.32, line 23: this is an odd set of citations. Lynch-Stieglitz et al. (2007) is spot on, but

Hain et al. did not present observations, and Guihou et al., is mainly about the glacial inception. There is a wealth of arguments for shoaled NADW during the LGM based on proxy data going back to Duplessy, Curry, Boyle etc. . .

P.33, line 4: ". . .which would be expected to increase. . ."

P.33, line 7:". . .in the Atlantic Ocean during the last glacial. . ."

P.33, line 16: please evaluate the plausibility of post depositional dissolution removing the evidence everywhere, otherwise it seems like a throw-away comment.

P.34, line 9: "even more counterintuitive" . . .than what?

P.34, line 14: CO32- is only approximately equal to ALK-DIC. Also, throughout the manuscript carbonate ion concentrations should be referred to as [CO32-], as was suggested by the reviewers I think.

P.35, line 6: would it be more precise to refer to changes in shallow/deep carbon partitioning, or changes in deep sea respired carbon storage?

P.35, line 10: ". . .are of the same order as the [observed. . . simulated??] CO32- excursions..."

P.35, line 12: an alternative suggestion: "The discrepancy between inferred and observed [CO32-] changes across the last glacial cycle emphasises the likely importance of changes in the distribution of carbon within the ocean. . ."

P.36, line 7: As proposed by a reviewer, please avoid starting a new sentence and subsection with "Yet. . .". Indeed, it might be better to write: "Even if the climatic impact. . ."

P.37, line 27: ". . .during the Holocene. . . compared to the glacial.."

P.38, line 26: ". . .unquestionably would have increased the 'active' carbon..." Incidentally, I wonder, is there a 'non-active' ALK inventory? A question of definition no doubt.

P.38, line 27: ". . .followed by the reduction of these inventories. . . and rapid shelf burial

resumed."

P.41, line 2: "Each record may be biased relative to its local"? What does this mean?

---

## Author Response (AR1)

*Dear Dr. Skinner,*
*Thank you very much for the effort you put in this manuscript, we really appreciate it. You will find below the description of the changes we made according to your comments.*

P.4, line 3: some of these references refer to peat/permafrost carbon, yet you define these reservoirs as being part of the 'active' carbon pool, rather than the 'geological' carbon pool. Please reconcile, or clarify.

*We removed the references for peat/permafrost*

P.4, line 13: Perhaps: ": : :allows for some first order estimates of changes in: : : These estimates suggest that large changes in both carbon and: : :."

*Modified accordingly*

P.4, line 18: I think at this point, given the length and complexity of the manuscript, it might be useful to provide a sort of 'key' to what follows, e.g. " In section 2 we provide an overview of: : :. In section 3 we present our reconstruction... and methods: : : In section 4 we describe some modelling scenarios: : : etc: : :"

*done*

P.4, line 24: "These quantitative constraints: : : past changes in the global carbon and alkalinity budgets."

*corrected*

P.5, line23: shells and shell fragments? Should we describe coccoliths as shells?

*True, the use of « shell » for cocco's isn't appropriate. reformulated.*

P.5, line 26: ": : :with increasing pressure and therefore water depth."

*modified*

P.9, line 3: In general, when referring to carbon, please be specific as to organic carbon, carbonate, or both.

*Modified into « Corg » although note that this section is only about organic carbon as stated in the title of the section. Note we have also changed 'carbonate' to $CaCO_3$ throughout.*

P.10, line 10: Are you saying that the most recent rapid sea level drop was at the start of the last deglaciation? Please correct or clarify.

*True, the sentence was unclear : reformulated it and added references.*

P.10, line 13: there has been a recent proposal that this value might be more positive, closer to arc volcanoes at -3 permil (Mason et al., 2017). Is this worth noting? Does it make a difference to your calculations?

*Thanks for pointing this out - indeed, changing the d13c of volcanic emission from -5 to -3 should reduce the amplitude of the changes due to volcanic emission on the active carbon pool. Although, this has already been partly explored in {Roth, 2012 #2709} in their exploration of uncertainty in the d13C of emissions, and the effect of volcanic emission remains low compared organic carbon fluxes (see figure 10). Note that we don't discuss the isotopic impact of any kind of deglacial pulse as it would differ depending on the nature of such a pulse. I nevertheless added a reference to {Roth, 2012 #2709} in the corresponding section 4.4.*

P.11, line 1: pleas change to 'the glacial' or 'glacials'.
P.11, line 2: ditto: : :

*Corrected*

P.11, line 4: Please break up the sentence. E.g. ": : :although this too has been debated (von Blanckenburg et al., 2015). It is also possible that: : : lower temperature, offsetting any increase in physical erosion and allowing global erosion to remain relatively stable: : :"

*Done*

P.11, line 6: Note also some new perspectives on the glacial weathering and the temperature feedback (e.g. Torres et al., PNAS 2017).

*Thanks, reference was inserted.*

P.12, line 10: I suggest to formulate this as a 'bulleted' list, allowing the intervening sentences: 1): : :; 2): : :; and 3)

*Done*

P.12, line 21: assumes.

*Corrected*

P.12, line 22: replace nor with or. I think you can cut "which would be expected to alter etc: : :" Perhaps instead at this stage it is worth making a brief note on the complicating effects of a world always out of equilibrium, contra the approach adopted here?

*We added some consideration on the non equilibrium issue.*

P.14, line 19: ": : :(no change), while: : :."

*Corrected*

P.21, lines 10-21: this whole paragraph is very unclear. Please have a go at clarifying what is meant.

*We reformulated this paragraph for more clarity.*

P.22, line 9: On P.18, line 6 it is stated that "we chose to correct for compaction.. by assuming.. global bulk burial rate was equal during the late Holocene and MIS5e: : :"
It seems wrong to then 'report' that mean deep sea carbonate burial was not different form modern during MIS5e, or have I missed something? Please correct or clarify.

*We only assume equivalent Bulk MAR for Holocene and MIS5. No assumption is made for the carbonate content. We clarified this issue in the text: "In part, the similarity of CaCO₃ burial between MIS 5 and the Holocene reflects our assumption that* bulk *sediment MAR during MIS 5 and the Holocene were similar, which underlies our detrending. However, MIS5 and the Holocene are also indistinguishable in the burial flux reconstruction estimated from CaCO₃ concentration alone (Figure 5D), in which MIS 4-2 also stands out as having significantly reduced burial. "*

P.23, line 3: The phrasing here is not optimal I think; please consider re-phrasing.

*This section was rephrased*

P.23: Here is a more substantial issue: the 'model' is not adequately described. Indeed, one of the reviewers asked for equations, and this has not been addressed as far as I can tell. The equation that is provided simply aims at describing the general notion of mass balance, though it needs to be corrected since a rate of change of concentration [C] cannot be equal to the difference between two concentrations; different symbols/nomenclature must be used to distinguish [C] from d[C]/dt, and to include the relevance of reservoir size etc... Furthermore, it is important that the air-sea partitioning scheme and shallow/deep partitioning scheme are fully described, and that equations for the rate of change of e.g. [C] in the atmosphere, shallow and deep ocean etc: : : be provided.

*We agree that the formulation we used was inappropriate to correctly describe the model simulation. Here we used the very basic and well known 3 box model described in {Sarmiento, 1984 #3094} then in {Toggweiler, 1999 #3210} with the exact same parameters used by {Toggweiler, 1999 #3210}. We think that providing the detailed equation without further discussion would obfuscate our message with no added value. Furthermore, the 3-box model really doesn't do much – the inventory results are essentially the outcome of the mass balance calculations, and the use of the 3-box model is a legacy from an early version of the paper that reported changes in atmospheric CO₂. Thus we have reformulated this section to provide all information required by referring to the original publications clearly, and also mention the fact that the impact of the 3-box model on the results is negligible.*

P.23, line 16: please consider rephrasing, e.g.: ": : :we have not considered variable erosion fluxes, due to a paucity of consistent observational constraints on these: : :"

*reformulated*

P.24: I think that at this stage in the manuscript it would be useful to provide a short

overview of the various scenarios that are considered and why (i.e. of what is to come in the next subsections). E.g. "Next we consider X different model scenarios where we calculate XYZ, on the basis of assumed inputs XYZ etc etc: : :. We label/name each of these scenarios KJH, as listed in table 3: : : Our goal with these scenarios is to: : : "
P.25: Is it possible to clearly link each of these sub-sections to the relevant model scenarios, e.g. in the title, or in the first few lines? I think it would be helpful.

*Done. ; I also changed the table 3 to make it more clear (bold characters for variable flux over times)*

P.25, line 12: Please add another word or two to this line to specify why the minimum is a robust outcome of the CaCO3 burial fluxes. The minimum is close to the minimum in carbonate/ALK output, which would be counter-intuitive given constant input fluxes, though I suspect that the minimum in ALK in fact occurs just before the minimum in carbonate burial (MIS4) and results from the sustained high/peak outputs leading up to the MIS5/4 transition. Please clarify.

*I modified the text : what we meant to be robust here is not the timing of the changes, but the general increase in alk from MIS4 to MIS2 which is the direct consequence of reduced burial during that period.*
*A minimum or a maximum in the alk occurs when the net alkalinity budget is equal to 0 (not shown in figures). Depending on the absolute value and shape of the removal flux, the period when the net alkalinity flux =0 can shift slightly. The amplitude of the minimum/maximum depends on the integral of the net alkalinity budget.*

Fig. 7: In this figure (and the others) please identify the vertical shaded bars. Also, it might be useful to clarify that 'net C source' is the net flux of carbon from the geological to the active carbon reservoirs, as defined here: : : if indeed this is correct.

*Correct and done!*

Fig. 7, caption: I think it would be clearer to write: "A. Variable deep sea CaCO3.. B. Variable deep dea.." etc.. The same goes for the other figures.

*Done*

P.26, line 3: ": : :experiments, the mean.."

*Done*

P.26, line 4: is it worth noting by how much 500PgC exceeds the modern flux, and that this would imply a downward drift in atmospheric CO2 across a glacial cycle if it was not actually realised?

*Done.*
*I added a reference to figure 1 and quote the value.*

P.32, line 8: This is a sentence fragment; please amend. E.g. "Although the preserved: : : (appendix A), our broad spatial: : :."

*Modified accordingly*

P.32, line 12: remove 'then': : : Though I think this could be stated much more clearly with a different construction: e.g. "Absolute changes in CaCO3 burial were obtained by scaling/multiplying reconstructed relative changes in CaCO3 MAR to/by the modern MAR. Uncertainties were estimated: : :"

*Modified accordingly*

P32, line 17: is 'very large' necessary?

*Removed*

P.32, line 18: again, was it not assumed a priori that MIS5 CaCO3 burial was the same as modern/late Holocene? Hence, for example: "on the assumption that it was the same during MIS5 and the late Holocene, CaCO3 burial dropped to the lowest rates: : :".

*As mentioned above, the long term trend correction is based only on the bulk MAR. Thus, we did not actually assume equal CaCO3 MAR for holocene and MIS5. I have now detailed this on p24.*

P.32, line 23: this is an odd set of citations. Lynch-Stieglitz et al. (2007) is spot on, but Hain et al. did not present observations, and Guihou et al., is mainly about the glacial inception. There is a wealth of arguments for shoaled NADW during the LGM based on proxy data going back to Duplessy, Curry, Boyle etc: : :

*modified accordingly*

P.33, line 4: ": : :which would be expected to increase: : :"

*corrected*

P.33, line 7:": : :in the Atlantic Ocean during the last glacial: : :"

*corrected*

P.33, line 16: please evaluate the plausibility of post depositional dissolution removing the evidence everywhere, otherwise it seems like a throw-away comment.

*removed*

P.34, line 9: "even more counterintuitive" : : :than what?

*removed*

P.34, line 14: CO32- is only approximately equal to ALK-DIC. Also, throughout the manuscript carbonate ion concentrations should be referred to as [CO32-], as was suggested by the reviewers I think.

*corrected*

P.35, line 6: would it be more precise to refer to changes in shallow/deep carbon partitioning, or changes in deep sea respired carbon storage?

*modified*

P.35, line 10: ": : :are of the same order as the [observed: : : simulated??] CO32- excursions..."

*corrected*

P.35, line 12: an alternative suggestion: "The discrepancy between inferred and observed [CO32-] changes across the last glacial cycle emphasises the likely importance of changes in the distribution of carbon within the ocean: : :"

*modified accordingly*

P.36, line 7: As proposed by a reviewer, please avoid starting a new sentence and subsection with "Yet: : :". Indeed, it might be better to write: "Even if the climatic impact: : :"

*modified*

P.37, line 27: ": : :during the Holocene: : : compared to the glacial.."

*Modified*

P.38, line 26: ": : :unquestionably would have increased the 'active' carbon..." Incidentally, I wonder, is there a 'non-active' ALK inventory? A question of definition no doubt.

*True, I changed the formulation to avoid any confusion.*

P.38, line 27: ": : :followed by the reduction of these inventories: : : and rapid shelf burial resumed."

*modified*

P.41, line 2: "Each record may be biased relative to its local"? What does this mean?

*We reformulated the sentence for more clarity.*

*Please note that, in addition to the changes noted above, we have made a number of further text changes throughout the manuscript, in order to improve clarity and readability. None of these makes a substantive change in the results, interpretations or conclusions.*

[revised manuscript text omitted]

olivier cartapanis 12/10/y 12:28

olivier cartapanis 12/10/y 12:28

olivier cartapanis 12/10/y 12:28

olivier cartapanis 12/10/y 12:28

olivier cartapanis 12/10/y 12:28

olivier cartapanis 12/10/y 12:28

olivier cartapanis 12/10/y 12:28

olivier cartapanis 12/10/y 12:28

olivier cartapanis 12/10/y 12:28

olivier cartapanis 12/10/y 12:28

olivier cartapanis 12/10/y 12:28

olivier cartapanis 12/10/y 12:28

olivier cartapanis 12/10/y 12:28

olivier cartapanis 12/10/y 12:28

olivier cartapanis 12/10/y 12:28

olivier cartapanis 12/10/y 12:28

olivier cartapanis 12/10/y 12:28

olivier cartapanis 12/10/y 12:28

olivier cartapanis 12/10/y 12:28

reconstruction of global deep-sea burial of $C_{org}$ was reported (Cartapanis et al., 2016), providing a record for one of these fluxes. Here, we present a new reconstruction of global deep sea (deeper than ≈ 200 m) burial of $CaCO_3$, completing the deep sea budgets of carbon and alkalinity burial. This reconstruction constrains the net result of carbonate compensation and allows for first order estimates of changes in carbon and alkalinity inventories to be illustrated. These estimates suggest that large changes in both carbon and and alkalinity inventories would have been driven by changes in burial, and highlight the potentially important role of variations in shelf burial fluxes, which remain poorly constrained.

The organization of this paper is as follows. In section 2, we provide an overview of natural sinks and sources of carbon in the climatic system. Section 3 presents the global deep sea $CaCO_3$ burial reconstruction, including the analytical strategy used and a detailed evaluation of the role of bulk mass accumulation rate and $CaCO_3$ concentrations in the reconstruction. In section 4 we explore the implication of the carbonate burial reconstruction on carbon and alkalinity inventories using simple modeling scenarios, and also include feasible changes in shelf burial and geological carbon release. Finally, we discuss the implications of our results in section 5.

**2. Natural sinks and sources of carbon in the climatic system**

[revised manuscript text omitted]